# Uremic toxin indoxyl sulfate induces trained immunity via the AhR-dependent arachidonic acid pathway in end-stage renal disease (ESRD)

Hee Young Kim[1,2]*†, Yeon Jun Kang[3], Dong Hyun Kim[3], Jiyeon Jang[3], Su Jeong Lee[3], Gwanghun Kim[4], Hee Byung Koh[5], Ye Eun Ko[5], Hyun Mu Shin[4,6], Hajeong Lee[7], Tae-Hyun Yoo[8], Won-Woo Lee[1,2,3,9]*†

[1]Department of Microbiology and Immunology, Seoul National University College of Medicine, Seoul, Republic of Korea; [2]Institute of Endemic Diseases, Seoul National University Medical Research Center, Seoul National University College of Medicine, Seoul, Republic of Korea; [3]Laboratory of Autoimmunity and Inflammation (LAI), Department of Biomedical Sciences, and BK21Plus Biomedical Science Project, Seoul National University College of Medicine, Seoul, Republic of Korea; [4]Department of Biomedical Sciences, College of Medicine and BK21Plus Biomedical Science Project, Seoul National University College of Medicine, Seoul, Republic of Korea; [5]Department of Internal Medicine, College of Medicine, Yonsei University, Seoul, Republic of Korea; [6]Wide River Institute of Immunology, Seoul National University, Hongcheon, Republic of Korea; [7]Division of Nephrology, Department of Internal Medicine, Seoul National University Hospital, Seoul, Republic of Korea; [8]Division of Nephrology, Department of Internal Medicine, Yonsei University College of Medicine, Seoul, Republic of Korea; [9]Seoul National University Cancer Research Institute; Ischemic/Hypoxic Disease Institute, Seoul National University Medical Research Center, Seoul National University Hospital Biomedical Research Institute, Seoul, Republic of Korea

*For correspondence:
hyk0801@hotmail.com (HYK);
wonwoolee@snu.ac.kr (WWL)

†These authors contributed equally to this work

Competing interest: The authors declare that no competing interests exist.

**Abstract** Trained immunity is the long-term functional reprogramming of innate immune cells, which results in altered responses toward a secondary challenge. Despite indoxyl sulfate (IS) being a potent stimulus associated with chronic kidney disease (CKD)-related inflammation, its impact on trained immunity has not been explored. Here, we demonstrate that IS induces trained immunity in monocytes via epigenetic and metabolic reprogramming, resulting in augmented cytokine production. Mechanistically, the aryl hydrocarbon receptor (AhR) contributes to IS-trained immunity by enhancing the expression of arachidonic acid (AA) metabolism-related genes such as arachidonate 5-lipoxygenase (ALOX5) and ALOX5 activating protein (ALOX5AP). Inhibition of AhR during IS training suppresses the induction of IS-trained immunity. Monocytes from end-stage renal disease (ESRD) patients have increased ALOX5 expression and after 6 days training, they exhibit enhanced TNF-α and IL-6 production to lipopolysaccharide (LPS). Furthermore, healthy control-derived monocytes trained with uremic sera from ESRD patients exhibit increased production of TNF-α and IL-6. Consistently, IS-trained mice and their splenic myeloid cells had increased production of TNF-α after in vivo and ex vivo LPS stimulation compared to that of control mice. These results provide insight into the role of IS in the induction of trained immunity, which is critical during inflammatory immune responses in CKD patients.

## eLife assessment

The authors expand the concept of a new layer to training immunity, which is currently being high-lighted by several colleagues in the field. The work provides **important** hints to understand end-stage renal disease. Overall, the rational approach leads to experimental results that are **solid**.

## Introduction

Over the last decade, a large body of evidence has demonstrated that innate cells can build up immunological memory resulting in enhanced responsiveness to subsequent stimulation, a phenomenon termed trained immunity (*Netea et al., 2011*). Compared with classical epitope-specific adaptive immunological memory based on an antigen-receptor, trained immunity of monocytes and macrophages is the long-term functional reprogramming elicited by an initial primary insult, mainly pathogen-associated molecular patterns (PAMPs), which leads to an altered response towards a subsequent, unrelated secondary insult after the return to a homeostatic state (*Netea et al., 2016*; *Netea et al., 2020*). It has been well demonstrated that exposure of monocytes or macrophages to *Candida albicans*, fungal cell wall component β-glucan, or Bacille Calmette-Guérin (BCG) vaccine enhances their subsequent responses to unrelated pathogens or pathogen components such as lipopolysaccharide (LPS) (*Arts et al., 2016b*; *Bekkering et al., 2018*). The induction of trained immunity is associated with the interaction of epigenetic modifications and metabolic rewiring, which can last for prolonged periods of time (*Netea et al., 2016*; *Netea et al., 2020*; *Bekkering et al., 2018*; *Cheng et al., 2014*; *Saeed et al., 2014*; *Christ et al., 2018*). Mechanistically, certain metabolites derived from the upregulation of different metabolic pathways triggered by primary insult can influence enzymes involved in remodeling the epigenetic landscape of cells. This leads to specific changes in epigenetic histone markers, such as histone 3 lysine 4 trimethylation (H3K4me3) or histone 3 lysine 27 acetylation (H3K27ac), which regulate genes resulting in a more rapid and stronger response upon a subsequent, unrelated secondary insult (*Netea et al., 2016*; *Netea et al., 2020*; *Arts et al., 2016b*; *Saeed et al., 2014*). In addition, it has been recently reported that long non-coding RNAs induce epigenetic reprogramming via the histone methyltransferase, MLL1. Subsequently, transcription factors such as Runx1 regulate the induction of proinflammatory cytokines following the secondary insult (*Fanucchi et al., 2019*; *Edgar et al., 2021*; *Jentho et al., 2021*).

Many studies have provided evidence that trained immunity likely evolved as a beneficial process for non-specific protection from future secondary infections (*Netea et al., 2020*). However, it has also been suggested that augmented immune responses resulting from trained immunity is potentially relevant to deleterious outcomes in immune-mediated and chronic inflammatory diseases such as autoimmune diseases, allergy, and atherosclerosis (*Christ et al., 2018*; *Edgar et al., 2021*; *Bekkering et al., 2014*; *van der Valk et al., 2016*; *Arts et al., 2018*; *Mulder et al., 2019*). Thus, although most studies have focused on the ability of exogenous microbial insults to induce trained immunity, it is also conceivable that sterile inflammatory insults can evoke trained immunity. In support of this idea, oxLDL, lipoprotein a (Lpa), uric acid, hyperglycemia, and the Western diet have all been recently identified as endogenous sterile insults that induce trained immunity in human monocytes via epigenetic reprogramming (*Christ et al., 2018*; *Edgar et al., 2021*; *Bekkering et al., 2014*; *van der Valk et al., 2016*; *Cabău et al., 2020*). Thus, it is tempting to speculate that many endogenous insults that cause chronic inflammatory conditions may be involved in the induction of trained immunity in human monocytes and macrophages.

Chronic kidney disease (CKD) is recognized as a major non-communicable disease with increasing worldwide prevalence (*Chen et al., 2019*; *Couser et al., 2011*). Loss of renal function in CKD patients causes the accumulation of over 100 uremic toxins, which are closely associated with cardiovascular risk and mortality due to their ability to generate oxidative stress and a proinflammatory cytokine milieu (*Vanholder et al., 2003*). Reflecting this, cardiovascular disease (CVD) is a leading cause of death among patients with end-stage renal disease (ESRD; *Kato et al., 2008*). Indoxyl sulfate (IS) is a major uremic toxin derived from dietary tryptophan via fermentation of gut microbiota (*Kim et al., 2017*). Since it is poorly cleared by hemodialysis, IS is one of the uremic toxins present at higher than normal concentrations in the serum of CKD patients (*Duranton et al., 2012*; *Lim et al., 2021*) and is associated with the progression of CKD and the development of CKD-related complications such as CVD (*Gao and Liu, 2017*). We and others have shown that IS promotes the production of

proinflammatory cytokines such as TNF-α and IL-1β by monocytes and macrophages through aryl hydrocarbon receptor (AhR) signaling and organic anion transporting polypeptides 2B1 (OATP2B1)-Dll4-Notch Signaling (*Kim et al., 2017*; *Kim et al., 2019*; *Nakano et al., 2019*), suggesting a role of IS as an endogenous inflammatory insult in monocytes and macrophages. Moreover, pretreatment with IS greatly increases TNF-α production by human macrophages in response to a low dose of LPS (*Kim et al., 2019*). Despite the function of IS as an endogenous inflammatory insult in monocytes and macrophages, little is known with regard to whether IS induces trained immunity. Thus, we investigated whether exposure to IS triggers trained immunity in an in vitro human monocyte model and an in vivo mouse model, as well as the mechanisms involved in IS-induced trained immunity. Our data show that IS triggers trained immunity in human monocytes/macrophages via AhR-dependent alteration of the arachidonic acid (AA) pathway, epigenetic modifications, and metabolic rewiring. Thus, this suggests IS plays a critical role in the initiation of inflammatory immune responses in patients with CKD.

## Results

### IS induces trained immunity in human monocytes

To explore whether exposure to IS is involved in the induction of trained immunity in human monocytes, an in vitro model of trained immunity was applied as previously reported by the Netea group (*Bekkering et al., 2016*). Freshly isolated human CD14$^+$ monocytes were preincubated for 24 hr with or without IS and, after a subsequent 5-day culture in human serum, restimulated with LPS or Pam3cys for final 24 hr (*Figure 1A*). Preincubation of monocytes with IS led to enhanced production of TNF-α, a major monocyte/macrophage-derived inflammatory cytokine, upon LPS stimulation. Since 10 ng/ml of LPS significantly increased both TNF-α and IL-6 secretion in IS-trained macrophages (*Figure 1B*), we used this concentration of LPS in subsequent experiments. A clinically relevant concentration of IS in severe CKD has reported the range from 0.5 to 1.0 mmol/L (*Vanholder et al., 2003*). The preincubation effect of IS on cytokine production was observed at a concentration as low as 250 µM, which is the average IS concentration in patients with ESRD in our cohort (*Figure 1C*; *Kim et al., 2019*). Unlike IS, preincubation with other protein-bound uremic toxins (PBUTs), such as *p*-cresyl sulfate (PCS), hippuric acid (HA), indole 3-acetic acid (IAA), and kynurenic acid (KA), did not cause increased secretion of TNF-α or IL-6 in response to LPS stimulation (*Figure 1—figure supplement 1A*). In addition, there was no obvious effect on cell viability following pre-incubation of macrophages with 1000 µM of IS after a subsequent 5-day culture in human serum or after LPS stimulation (*Figure 1—figure supplement 1B*). We also found that the enhanced cytokine production of IS-trained macrophages was not attributable to potassium derived from IS potassium salt (*Figure 1—figure supplement 1C*). Moreover, the increased TNF-α and IL-6 production in IS-trained macrophages was not limited to LPS stimulation, as similar phenomena were observed following stimulation with Pam3cys, a TLR1/2 agonist (*Figure 1D*). β-glucan-pretreated macrophages exhibit a prototypic feature of trained immunity, characterized by enhanced production of inflammatory cytokines upon restimulation with heterologous stimuli, LPS or Pam3cys (*Cheng et al., 2014*; *Bekkering et al., 2016*). As seen in *Figure 1—figure supplement 1D*, the level of TNF-α secreted by IS-trained macrophages was comparable with that secreted by β-glucan-trained macrophages, although β-glucan had a more potent effect on IL-6 production than did IS, suggesting that IS plays a role in the induction of trained immunity of human monocytes. Furthermore, alongside elevated TNF-α and IL-6 expressions, there was a notable increase in the mRNA expression of *IL-1β* and *MCP-1* (*CCL2*) observed in IS-trained macrophages, concomitant with a significant reduction in *IL-10*, a cytokine known for its anti-inflammatory properties, within the same cellular context (*Figure 1E*). Correspondingly, alterations in their protein levels mirrored the observed mRNA expressions (*Figure 1—figure supplement 1E*). Circulating monocytes have been identified as a major immune cell subset that responds to IS in the serum of ESRD patients (*Kim et al., 2017*; *Kim et al., 2019*). To examine whether uremic serum induces trained immunity of monocytes/macrophages, pooled sera from ESRD patients (184±44 µM of average IS level) or from healthy controls (HCs) were used to treat monocytes isolated from HCs for 24 hr at 30% (v/v), followed by training for 5 days (*Figure 1F*). Training with pooled uremic serum of ESRD patients increased the production of TNF-α and IL-6 upon re-stimulation with LPS compared to monocytes treated with the pooled sera of HCs (*Figure 1G–I*). In addition, expression of *IL-1β* and *MCP-1* mRNA was also augmented by training

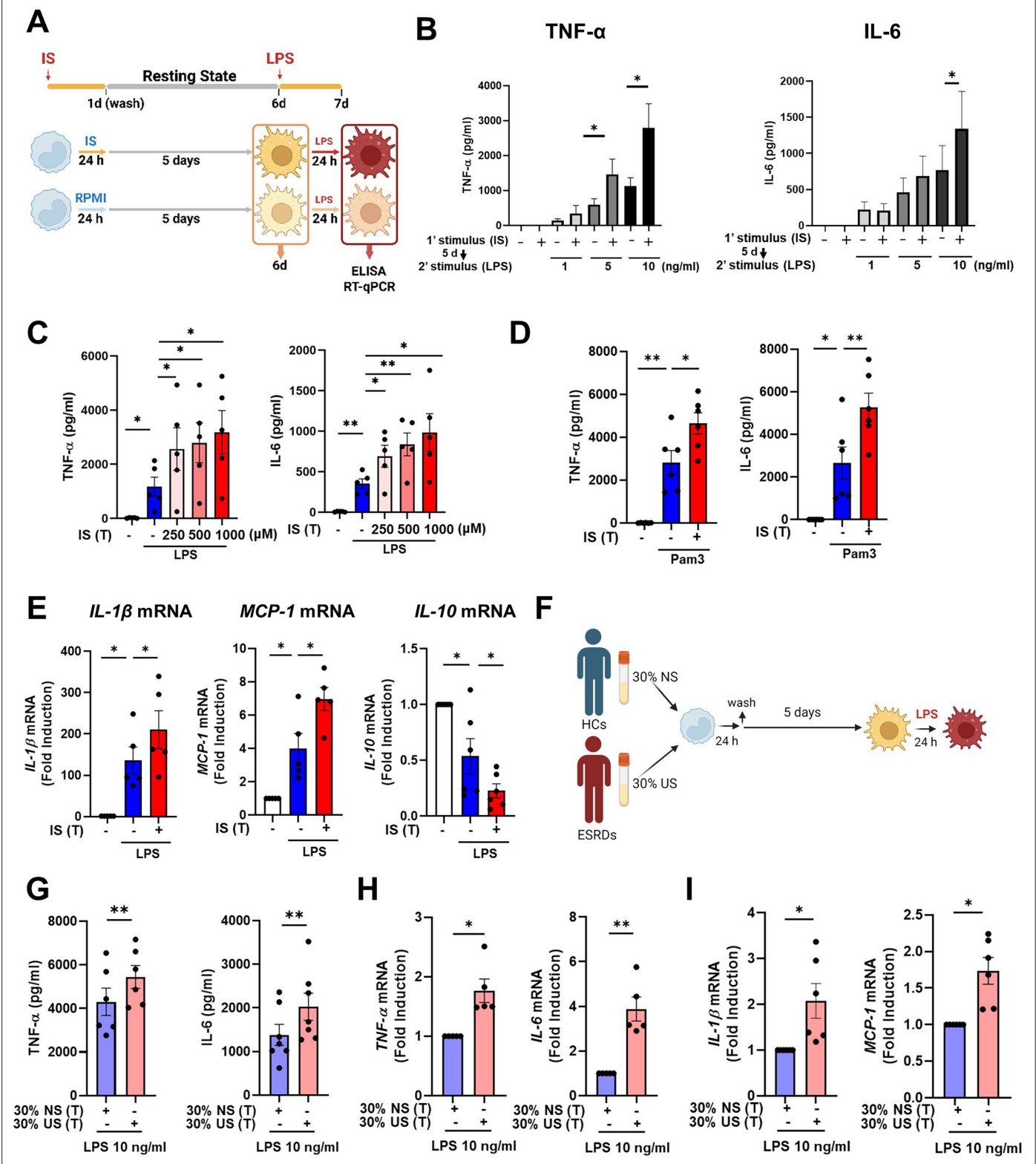

**Figure 1.** Indoxyl sulfate (IS) induces trained immunity in human monocytes. (**A**) Schematic of in vitro experimental model for innate trained immunity. (**B, C**) Human monocytes were treated with the indicated concentration of IS for 24 hr, followed by a subsequent 5-day culture in human serum. On day 6, the cells were restimulated with the indicated concentrations of lipopolysaccharide (LPS) for 24 hr. TNF-α and IL-6 proteins levels were quantified by enzyme-linked immunosorbent assay (ELISA). (**D**) After training with 1,000 μM IS, monocytes were restimulated with 10 μg/ml Pam3cys. TNF-α and IL-6 protein levels were quantified by ELISA. (**E**) After training with 1000 μM IS, monocytes were restimulated with 10 ng/ml LPS for 24 hr. The mRNA

*Figure 1 continued on next page*

*Figure 1 continued*

expression of *IL-1β*, *IL-10*, and *MCP-1* was analyzed by RT-qPCR. (**F**) In vitro experimental scheme of uremic serum-induced trained immunity. (**G–I**) The pooled normal serum (NS) from healthy controls (HCs) or uremic serum (US) from patients with end-stage renal disease (ESRD) were used for treatment of monocytes isolated from HCs for 24 hr at 30% (v/v) followed by resting for 5 days. After stimulation with LPS for 24 hr, TNF-α and IL-6 production were analyzed using ELISA (**G**) and RT-qPCR (**H**). After stimulation with LPS (10 ng/ml) for 24 hr, mRNA expression of *IL-1β* and *MCP-1* were determined by RT-qPCR (**I**). n=5 ~ 7. Bar graphs show the mean ± SEM. *=p < 0.05, and **=p < 0.01 by two-tailed paired *t*-test.

The online version of this article includes the following source data and figure supplement(s) for figure 1:

**Source data 1.** Raw data for *Figure 1B–E and G–I*.

**Figure supplement 1.** Indoxyl sulfate (IS) induces trained immunity in human monocytes.

**Figure supplement 1—source data 1.** Raw data for *Figure 1—figure supplement 1A–E*.

with the pooled uremic sera of ESRD patients (*Figure 1I*). These results suggest that IS induces trained immunity in human monocytes, characterized by the increased expression of proinflammatory cytokines TNF-α and IL-6 and reduced expression of anti-inflammatory IL-10 in response to secondary TLR stimulation.

## IS-induced trained immunity is regulated by metabolic rewiring

Metabolic rewiring is one of the most crucial processes regulating the trained immunity of monocytes and macrophages (*Brito et al., 2017*). Assessment of the metabolic profile of IS-trained macrophages on day 6 (prior to re-stimulation with LPS) showed that training with IS led to an enhanced extracellular acidification rate (ECAR) as a measure of lactate production, indicating increased glycolysis and glycolysis capacity (*Figure 2A and B*). Moreover, basal and maximal respiration and ATP production gauged by the oxygen consumption rate (OCR) were also increased compared to that of non-trained cells (*Figure 2C and D*). Enhanced glycolysis and glycolytic capacity in IS-trained cells remained higher even after re-stimulation with LPS (*Figure 2—figure supplement 1A and B*), implying that the IS-training effect on metabolic rewiring is sustained regardless of the secondary stimulation. To further examine whether the metabolic rewiring by IS-trained cells is linked to the regulation of trained immunity, 2-deoxy-d-glucose (2-DG), a general inhibitor of glycolysis, was added to monocytes before training with IS. 2-DG completely inhibited the augmented production of TNF-α and IL-6 in IS-trained macrophages in response to re-stimulation with LPS (*Figure 2E*). These data demonstrate that IS-trained immunity is linked to metabolic rewiring characterized by both enhanced glycolysis and augmented oxidative respiration.

## Epigenetic modifications control IS-induced trained immunity

The induction of trained immunity relies on two key, closely intertwined mechanisms, epigenetic modification and metabolic rewiring of innate immune cells (*Netea et al., 2016*; *Divangahi et al., 2021*; *Gourbal et al., 2018*). We next sought to determine whether increased expression of TNF-α and IL-6 is a result of epigenetic changes. To this end, chromatin modification of histone 3 trimethylation of lysine 4 (H3K4me3) at the promoter sites of *TNFA* and *IL6* was analyzed. Chromatin immunoprecipitation (ChIP)-qPCR data illustrate that IS-trained macrophages exhibit enhanced H3K4me3 of *TNFA* and *IL6* promoters by day 6 after treatment with 1000 µM of IS (*Figure 3A and B*). This reflects what was previously demonstrated in trained innate immune cells (*Arts et al., 2016b*; *Duranton et al., 2012*; *Stevens et al., 2013*). Moreover, IS-mediated enrichment of H3K4me3 was maintained even after secondary stimulation with LPS compared with non-trained cells (*Figure 3—figure supplement 1A and B*). When IS-trained macrophages were pretreated with 5'-methylthioadenosine (MTA), a non-selective methyltransferase inhibitor, their production of TNF-α and IL-6 upon LPS stimulation was reversed to baseline (*Figure 3C*), implying that IS-induced trained immunity is associated with epigenetic modification. To explore the potential regulation of IS-induced epigenetic modification by metabolic rewiring, we examined the enrichment of H3K4me3 at the promoters of *TNFA* and *IL6* subsequent to treatment with 2-DG (*Figure 3D*). Our findings suggest that metabolic rewiring influences epigenetic modification, implicating the participation of metabolites. Additionally, heightened enrichment of H3K4me3 at the promoter regions of *HK2* and *PFKP*, pivotal genes associated with

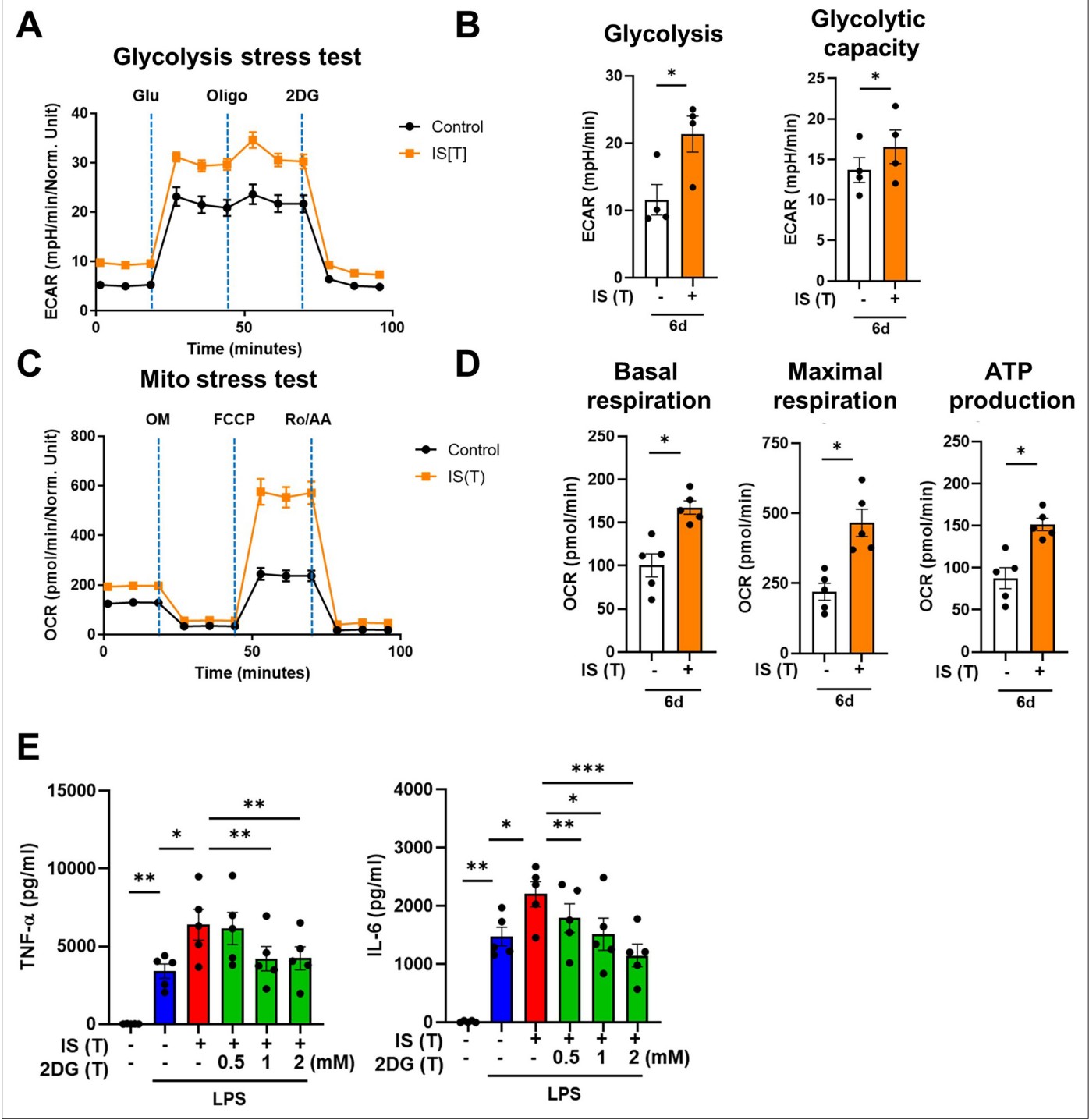

**Figure 2.** Indoxyl sulfate (IS)-induced trained immunity is linked to metabolic rewiring. Glycolysis and mitochondrial stress tests were conducted on IS (1000 μM)-trained macrophages (n=4 ~ 5) using the Seahorse XF-analyzer. (**A**) Extracellular acidification rate (ECAR) levels were measured after sequential treatment with glucose, oligomycin, and 2-DG. (**B**) Cellular glycolysis and glycolytic capacity were analyzed. (**C**) Oxygen consumption rate (OCR) levels were measured after sequential treatment with oligomycin, FCCP, and Rotenone/antimycin A (Ro/AA). (**D**) Basal respiration, maximal respiration, and ATP production were analyzed. (**E**) Monocytes were pretreated with 2-deoxy-d-glucose (2-DG), followed by IS-training for 6 days. Cells were restimulated with lipopolysaccharide (LPS) for 24 hr and TNF-α and IL-6 in supernatants were quantified by enzyme-linked immunosorbent assay (ELISA) (n=5). Bar graphs show the mean ± SEM. *=p < 0.05, **=p < 0.01, and ***=p < 0.001 by two-tailed paired *t*-test.

The online version of this article includes the following source data and figure supplement(s) for figure 2:

**Source data 1.** Raw data for *Figure 2B, D, and E*.

*Figure 2 continued on next page*

*Figure 2 continued*

**Figure supplement 1.** Indoxyl sulfate (IS)-induced trained immunity is linked to metabolic rewiring.

**Figure supplement 1—source data 1.** Raw data for *Figure 2—figure supplement 1B*.

glycolysis, was observed (*Figure 3—figure supplement 1C*). To further elucidate epigenetic modifications in IS-induced trained immunity, we performed a whole-genome assessment of the histone marker H3K4me3 by ChIP-sequencing (ChIP-Seq) in IS-trained cells on day 6. Among 7,136 peaks, 59 differentially upregulated peaks and 316 downregulated peaks were detected in IS-trained cells (*Figure 3E* and *Table 1*). To identify the biological processes affected in IS-mediated trained immunity, 59 upregulated peaks in IS-trained macrophages were analyzed through Gene Ontology (GO) analysis with Go biological process and the Reactome Gene Set. Activation of the innate immune response and positive regulation of the defense response were identified as major processes via Go biological process analysis. Further, genes involved in regulation of ornithine decarboxylase (ODC) and metabolism of polyamine were recognized as major gene sets via Reactome Gene Set analysis (*Figure 3F*). A genome browser snapshot showing H3K4me3 binding illustrates that H3K4me3 is elevated at the promoters of important target genes associated with activation of the innate immune response, such as *IFI16* (interferon-gamma inducible protein 16), *XRCC5* (X-ray repair cross-complementing 5), and *PQBP1* (polyglutamine binding protein 1) and genes linked to the regulation of ODC, such as *PSMA1* (proteasome 20 S subunit alpha 1), *PSMA3* (proteasome 20 S subunit alpha 3), and *OAZ3* (Ornithine Decarboxylase Antizyme 3, a protein that negatively regulates ODC activity) (*Figure 3G*; *Hardbower et al., 2017*). Additionally, differences in H3K4me3 enrichment patterns between the IS-training group and the control group were observed in *TNFA* and *IL6* (*Figure 3—figure supplement 1D and E*). Our results show that epigenetic modification of innate immune response-related genes contributes to the induction of IS-trained immunity in human monocytes.

## AhR, a potent endogenous receptor for IS, contributes to the induction of IS-trained immunity

Our previous study demonstrated that IS-induced TNF-α production in macrophages is regulated through a complex mechanism involving the interaction of NF-κB and SOCS2 with AhR (*Kim et al., 2019*). To explore the molecular mechanism underlying the regulation of IS-trained immunity, we investigated the role of AhR, a potent endogenous receptor for IS. Ligand-bound activated AhR is known to be immediately translocated into the nucleus and rapidly degraded (*Kim et al., 2019*; *Schroeder et al., 2010*; *Brito et al., 2017*; *Dou et al., 2018*). Immunoblot analysis depicted in *Figure 4A* reveals persistent nuclear translocation of IS-mediated AhR even on day 6 (prior to re-stimulation with LPS), which was entirely inhibited by GNF351 treatment, an AhR antagonist, on day 6. Inhibition of AhR by GNF351 during IS training suppressed the increase in production of TNF-α and IL-6 following LPS restimulation on day 6 in IS-trained cells (*Figure 4B and C*), implying that IS-mediated AhR activation may be involved in trained immunity. In addition to TNF-α and IL-6, enhancement of *IL-1β* and *MCP-1* mRNA expression in IS-trained cells was also completely inhibited, whereas decreased *IL-10* expression was completely reversed by GNF351 (*Figure 4D*). To confirm the regulatory role of AhR in trained immunity, we tested whether 6-Formylindolo[3,2-b]carbazole (FICZ), a tryptophan-derived agonist of AhR, also induced trained immunity in human monocytes. FICZ-pretreated monocytes exhibited augmented expression of TNF-α and IL-6 in response to secondary stimulation with LPS compared to non-trained cells (*Figure 4—figure supplement 1A*). Additionally, knockdown of AhR suppressed the expression of TNF-α and IL-6 in IS-trained cells (*Figure 4E*), underscoring the significant role of ligand-bound activated AhR in the trained immunity of human monocytes. We next examined whether inhibition of AhR with GNF351 influences epigenetic modification and metabolic rewiring. Our ChIP-qPCR assay showed that enrichment of H3K4m3 on *TNFA* and *IL6* promoters in IS-trained macrophages was inhibited by GNF351 (*Figure 4F*). Of note, assessment of the metabolic profile by measuring ECAR and OCR illustrates that GNF351 has no effect on metabolic rewiring, including enhanced glycolysis and mitochondrial respiration, in IS-trained cells on day 6 (*Figure 4—figure supplement 1*). This finding was corroborated by the immunoblotting data, which showed GNF351 had no inhibitory effect on IS-mediated enhancement of S6K activity, which is critical for inducing the aerobic glycolysis in

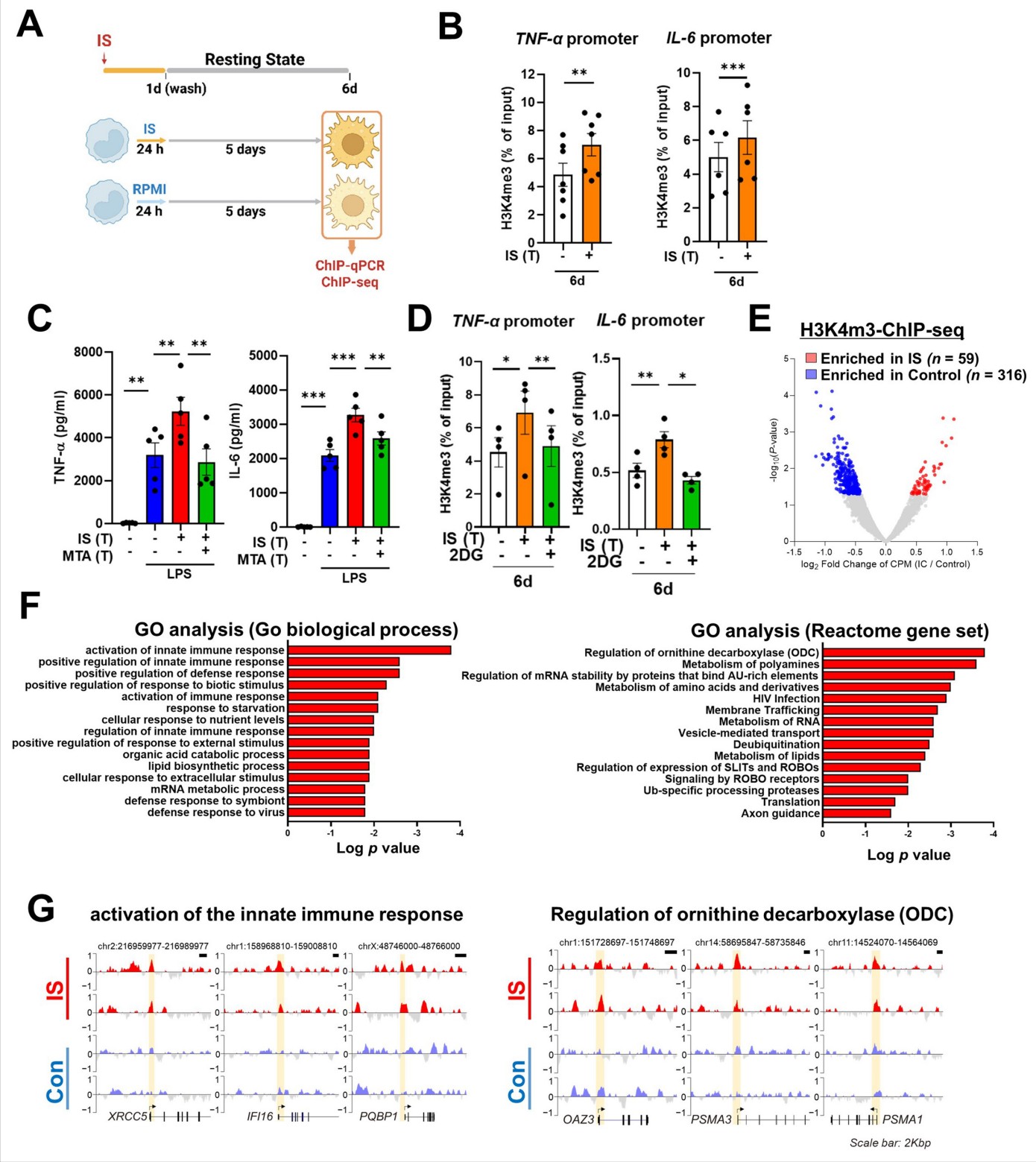

**Figure 3.** Indoxyl sulfate (IS)-induced trained immunity is accomplished through epigenetic modification. (**A**) Experimental scheme of chromatin immunoprecipitation (ChIP)-qPCR for IS (1000 μM)-trained macrophages. (**B**) On day 6 after IS-training, cells were fixed with 1% formaldehyde, lysed, and sonicated. A ChIP assay was performed using anti-H3K4me3 antibody and enrichment of H3K4me3 in the promoter site of *TNFA* (n=7) and *IL6* (n=6) loci was quantified by qPCR. 1% input was used as a normalization control. (**C**) Monocytes were pre-treated with 5'-methylthioadenosine (MTA, a non-selective methyltransferase inhibitor; 200 μM) and then were trained with IS for 6 days, followed by restimulation with lipopolysaccharide (LPS) for 24 hr.

*Figure 3 continued on next page*

*Figure 3 continued*

TNF-α and IL-6 proteins levels were quantified by enzyme-linked immunosorbent assay (ELISA) (n=4 ~ 5). (D) A ChIP assay was performed in IS-trained macrophages pre-treated with 2-deoxy-d-glucose (2-DG) (n=4). 2% input was used as a normalization control. (E) ChIP-sequencing (ChIP-Seq) analysis was performed with anti-H3K4me3 antibody on chromatin isolated at day 6 from IS-trained and control macrophages. Enriched peaks in ChIP-Seq on H3K4me3 are shown as a volcano plot. (FC >1.3, p<0.05) (F) Functional annotation of 59 upregulated differentially regulated peaks (DRPs) on H3K4me3 in IS-trained macrophages were analyzed by Gene Ontology (GO) analysis with Go biological pathway and Reactome gene sets (FC >1.3, p<0.05). (G) Screen shots of H3K4me3 modification in the promoter regions of *IFI16, XRCC5, PQBP1 PSMA1, PSMA3,* and *OAZ3*. *=p < 0.05, **=p < 0.01, and ***=p < 0.001 by two-tailed paired *t*-test.

The online version of this article includes the following source data and figure supplement(s) for figure 3:

**Source data 1.** Raw data for *Figure 3B–D*.

**Figure supplement 1.** Indoxyl sulfate (IS)-induced trained immunity is associated with epigenetic modification in human innate immune cells.

**Figure supplement 1—source data 1.** Raw data for *Figure 3—figure supplement 1B and C*.

human monocytes/macrophages (*Figure 4—figure supplement 1D*). Our findings suggest that IS-activated AhR is involved in regulating epigenetic modifications of IS-trained macrophages.

## AhR-dependent induction of the AA pathway is involved in IS-induced trained immunity

To explore which molecular mechanism is involved in the induction of IS-trained immunity, we performed RNA-sequencing (RNA-Seq) on day 6 (prior to restimulation with LPS) in IS-trained human macrophages. A total of 218 differentially expressed genes (DEGs), consisting of 71 upregulated and 147 downregulated genes, were identified in IS-trained macrophages compared to non-trained cells (*Figure 5A* and *Figure 5—figure supplement 1A*; FC ≥± 2, <0.05). GO analysis of these expression data using the Reactome Gene Set is displayed in *Figure 5B*. IS-trained macrophages had upregulated pathways including those involved in neutrophil degranulation, integrin cell surface interactions, extracellular matrix organization and AA metabolism, whereas pathways associated with kinesins, cell cycle, and the Gα(i) signaling pathway were downregulated (*Figure 5B*). Considering the key role of the AA pathway in many inflammatory disorders, we decided to focus on this pathway in the induction of trained immunity by IS. Our findings were supported by Gene Set Enrichment Analysis (GSEA) using Molecular Signatures Database (MsigDB), in which genes related to AA metabolism were enriched in IS-trained macrophages compared to non-trained cells and more importantly, upregulated expression of these genes was inhibited by treatment with GNF351 as illustrated by heatmap analysis of major genes related to AA metabolism (*Figure 5C and D* and *Figure 5—figure supplement 1G*). Among AA metabolism pathways, the leukotriene metabolic process, but not the cyclooxygenase (COX) pathway, was primarily involved in the induction of IS-mediated trained immunity (*Figure 5—figure supplement 1B*). Confirmatory RT-qPCR analysis on major AA metabolism-related genes was conducted using IS-trained macrophages obtained from independent, HCs (*Figure 5E*). The mRNA expression of arachidonate 5-lipoxygenase (*ALOX5*: also known as *5-LOX* or *5-LO*) and ALOX5 activating protein (*ALOX5AP*: also known as *FLAP*), the enzymes catalyzing AA into leukotrienes (a group of pro-inflammatory lipid mediators) (*Salina et al., 2020*; *Rådmark et al., 2007*; *Pernet et al., 2019*), was higher in IS-trained macrophages than non-trained cells. In addition, the mRNA expression of *LTB4R1* (also known as *BLT1*), a high-affinity receptor for leukotriene B4 (LTB4), was also upregulated. The augmented expression of these AA metabolism-related genes was repressed by GNF351 pretreatment as shown by changes in expression of *CYP1B1*, a typical AhR target gene. Thus, this suggests that the IS-activated AhR pathway is involved in enhanced AA-metabolism in IS-induced trained immunity. Immunoblot analysis validated the upregulation of ALOX5 and ALOX5AP expression in IS-trained immunity, which was subsequently inhibited by GNF351 at the protein level (*Figure 5F*). Furthermore, knockdown of AhR suppressed the IS-induced mRNA expression of *ALOX5, ALOX5AP,* and *LTB4R* on day 6 (*Figure 5G*). Treatment with FICZ, an AhR agonist known to induce trained immunity (*Figure 4—figure supplement 1A*), elicited increased expression of *ALOX5* and *ALOX5AP*, while treatment with KA, a major protein-bound uremic toxin that does not induce trained immunity

**Table 1.** The 59 differentially upregulated enriched peaks in indoxyl sulfate (IS)-trained cells at day 6.

| No. | Fold change (IS/Ctrl) | Symbol | p-Value | Chromosome | Start | End |
|-----|-----|-----|-----|-----|-----|-----|
| 1 | 2.17 | PTMA | 0.000 | chr2 | 232,572,225 | 232,572,892 |
| 2 | 2.13 | TAF9B | 0.001 | chrX | 77,394,594 | 77,395,229 |
| 3 | 1.99 | ULK1 | 0.002 | chr12 | 132,379,673 | 132,380,598 |
| 4 | 1.95 | HCN1 | 0.024 | chr5 | 46,391,617 | 46,393,029 |
| 5 | 1.92 | PRPF4B | 0.000 | chr6 | 4,018,154 | 4,019,021 |
| 6 | 1.90 | TPM2 | 0.002 | chr9 | 35,690,429 | 35,691,336 |
| 7 | 1.89 | PLCD1 | 0.008 | chr3 | 38,065,564 | 38,066,458 |
| 8 | 1.87 | ZXDA | 0.008 | chrX | 58,548,803 | 58,549,951 |
| 9 | 1.84 | CLCN5 | 0.010 | chrX | 49,683,035 | 49,684,086 |
| 10 | 1.74 | SCLY | 0.010 | chr2 | 238,968,675 | 238,969,378 |
| 11 | 1.74 | EXOSC5 | 0.015 | chr19 | 41,903,568 | 41,904,454 |
| 12 | 1.74 | ZCCHC24 | 0.012 | chr10 | 81,204,279 | 81,205,209 |
| 13 | 1.73 | NCAPG2 | 0.009 | chr7 | 158,497,836 | 158,498,379 |
| 14 | 1.66 | ZFP69B | 0.028 | chr1 | 40,889,909 | 40,890,810 |
| 15 | 1.65 | PIGP | 0.031 | chr21 | 38,442,782 | 38,443,698 |
| 16 | 1.63 | RPS12 | 0.023 | chr6 | 133,134,412 | 133,135,505 |
| 17 | 1.63 | FNBP1L | 0.027 | chr1 | 93,920,253 | 93,920,934 |
| 18 | 1.63 | KDSR | 0.036 | chr18 | 61,035,043 | 61,035,711 |
| 19 | 1.62 | MIR4436A | 0.030 | chr2 | 90,300,121 | 90,300,965 |
| 20 | 1.61 | GPSM3 | 0.033 | chr6 | 32,163,701 | 32,164,335 |
| 21 | 1.60 | FKBP11 | 0.031 | chr12 | 49,318,763 | 49,319,548 |
| 22 | 1.60 | PRKAG2 | 0.041 | chr7 | 151,605,461 | 151,606,648 |
| 23 | 1.59 | IFI16 | 0.036 | chr1 | 158,979,768 | 158,981,235 |
| 24 | 1.58 | MAOA | 0.050 | chrX | 43,514,253 | 43,514,969 |
| 25 | 1.58 | XRCC5 | 0.022 | chr2 | 216,974,068 | 216,974,992 |
| 26 | 1.58 | PQBP1 | 0.015 | chrX | 48,754,482 | 48,755,683 |
| 27 | 1.58 | TSNARE1 | 0.034 | chr8 | 143,483,367 | 143,484,264 |
| 28 | 1.57 | ENOSF1 | 0.025 | chr18 | 711,957 | 712,856 |
| 29 | 1.56 | RAD23A | 0.007 | chr19 | 13,056,634 | 13,057,623 |
| 30 | 1.56 | ACTR3 | 0.032 | chr2 | 114,646,472 | 114,647,252 |
| 31 | 1.55 | C5orf51 | 0.023 | chr5 | 41,904,386 | 41,905,346 |
| 32 | 1.55 | UCHL1-AS1 | 0.020 | chr4 | 41,258,857 | 41,260,104 |
| 33 | 1.54 | EEPD1 | 0.038 | chr7 | 36,195,035 | 36,196,312 |
| 34 | 1.54 | ZNF585B | 0.048 | chr19 | 37,700,961 | 37,701,592 |
| 35 | 1.53 | PPA2 | 0.010 | chr4 | 106,394,085 | 106,395,366 |
| 36 | 1.52 | EIF1AX | 0.016 | chrX | 20,159,079 | 20,160,075 |
| 37 | 1.51 | CD53 | 0.027 | chr1 | 111,415,818 | 111,417,201 |
| 38 | 1.51 | NUDCD3 | 0.011 | chr7 | 44,529,338 | 44,530,500 |
| 39 | 1.49 | SPATA1 | 0.012 | chr1 | 84,970,305 | 84,971,951 |

*Table 1 continued on next page*

Table 1 continued

| No. | Fold change (IS/Ctrl) | Symbol | p-Value | Chromosome | Start | End |
|---|---|---|---|---|---|---|
| 40 | 1.48 | HSD17B11 | 0.019 | chr4 | 88,311,038 | 88,312,383 |
| 41 | 1.47 | VPS53 | 0.042 | chr17 | 497,350 | 499,088 |
| 42 | 1.47 | FLYWCH2 | 0.045 | chr16 | 2,932,908 | 2,933,882 |
| 43 | 1.47 | RBBP9 | 0.048 | chr20 | 18,476,929 | 18,478,060 |
| 44 | 1.46 | TNFRSF21 | 0.025 | chr6 | 47,276,461 | 47,277,774 |
| 45 | 1.45 | LOC101927974 | 0.029 | chr7 | 107,384,234 | 107,385,507 |
| 46 | 1.45 | OAZ3 | 0.041 | chr1 | 151,735,094 | 151,736,365 |
| 47 | 1.44 | TMEM219 | 0.026 | chr16 | 29,973,365 | 29,974,938 |
| 48 | 1.44 | CUTA | 0.047 | chr6 | 33,384,929 | 33,386,004 |
| 49 | 1.43 | PSMA3 | 0.023 | chr14 | 58,710,630 | 58,712,355 |
| 50 | 1.43 | PLRG1 | 0.046 | chr4 | 155,470,747 | 155,472,093 |
| 51 | 1.43 | PSMA1 | 0.050 | chr11 | 14,540,951 | 14,542,589 |
| 52 | 1.40 | TMEM131 | 0.036 | chr2 | 98,611,268 | 98,612,743 |
| 53 | 1.39 | RPUSD2 | 0.030 | chr15 | 40,861,310 | 40,862,576 |
| 54 | 1.39 | NEK4 | 0.043 | chr3 | 52,803,934 | 52,805,223 |
| 55 | 1.38 | TRIP11 | 0.035 | chr14 | 92,505,410 | 92,507,100 |
| 56 | 1.37 | ACAA1 | 0.046 | chr3 | 38,177,208 | 38,178,850 |
| 57 | 1.36 | ZNF212 | 0.050 | chr7 | 148,936,596 | 148,937,656 |
| 58 | 1.35 | LRRC8D | 0.044 | chr1 | 90,286,653 | 90,288,467 |
| 59 | 1.34 | PTPMT1 | 0.048 | chr11 | 47,586,495 | 47,588,063 |

(*Figure 1A*), did not result in elevation of these genes, thereby implying the significant role of the AhR-AA pathway in IS-trained immunity (*Figure 5—figure supplement 1C*).

We previously reported alterations in the transcriptome signature of ex vivo monocytes of ESRD patients (*Kim et al., 2020*). Comparison of the fold changes of RNA-Seq data in the present study and microarray data reported previously (GSE155326) revealed that the expression of *ALOX5* and *LTB4R1* is enhanced in IS-trained macrophages and ex vivo monocytes of ESRD patients (*Figure 5—figure supplement 1D*). To further investigate the roles of the AA metabolism pathway in IS-trained immunity, zileuton, an ALOX5 inhibitor, and U75302, a BLT1 receptor inhibitor were used during the induction of trained immunity by IS (*Figure 5H*, *Figure 5—figure supplement 1E and F*). We found that IS-induced TNF-α and IL-6 production were largely suppressed by both zileuton and U75302 (*Figure 5H* and *Figure 5—figure supplement 1F*). Additionally, knockdown of ALOX5 inhibited IS-induced expression of TNF-α and IL-6 (*Figure 5—figure supplement 1H*). In further exploration of the effects on epigenetic or metabolic reprogramming via the AA pathway, we conducted ChIP-qPCR assays and Western blot analyses following treatment with zileuton. Our results demonstrated that the enrichment of H3K4me3 on *TNFA* and *IL6* promoters in IS-trained macrophages was inhibited by zileuton, although phosphorylation of S6K remained unaffected (*Figure 5I* and *Figure 5—figure supplement 1I*). Thus, these findings suggest that AA metabolism plays a pivotal role in the induction of IS-trained immunity by serving as a crucial mediator between AhR signaling and epigenetic modification. We next tested whether training with uremic serum leads to increased expression of AA pathway-related genes within 6 days (prior to restimulation with LPS) as found in IS-trained macrophages. The expression of *ALOX5*, *ALOX5AP*, and *LTB4R1* mRNA was augmented by training with pooled uremic sera of ESRD patients compared with HCs (*Figure 5J*), and this augmented expression was maintained after re-stimulation with LPS (*Figure 5—figure supplement 1J*).

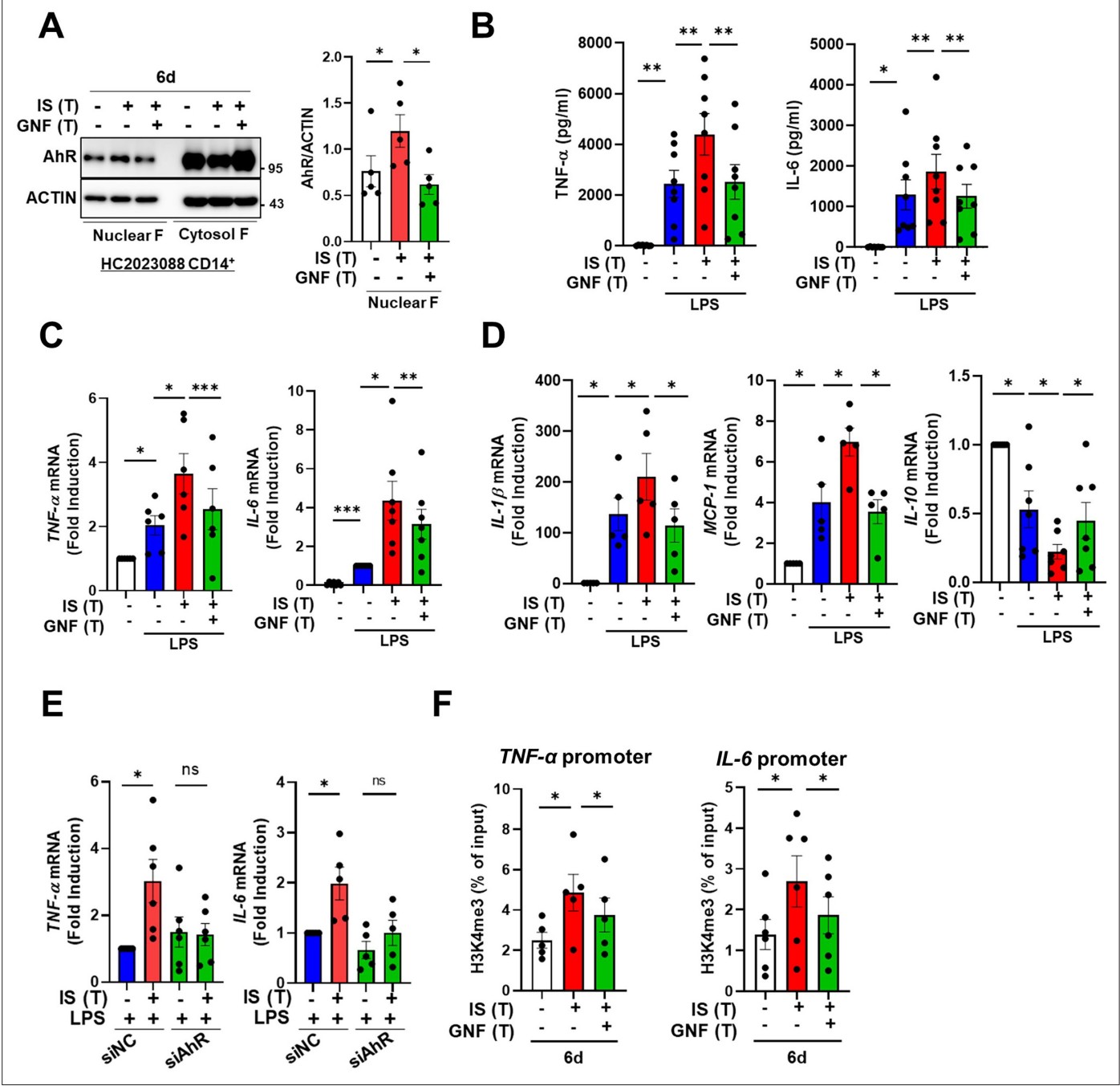

**Figure 4.** Indoxyl sulfate (IS)-induced trained immunity is regulated by aryl hydrocarbon receptor (AhR). Monocytes were pretreated with or without GNF351 (AhR antagonist; 10 μM) followed by IS (1000 μM)-training for 6 days. (**A**) On day 6, nuclear and cytosol fraction were prepared and immunoblotted for AhR protein. Band intensity in immunoblots was quantified by densitometry. β-ACTIN was used as a normalization control. (**B–D**) On day 6, IS-trained cells with or without GNF351 were restimulated with lipopolysaccharide (LPS) (10 ng/ml), for 24 hr. TNF-α and IL-6 in supernatants were quantified by enzyme-linked immunosorbent assay (ELISA) (**B**). Expression of TNF-α and IL-6 (**C**) and *IL-1β, MCP-1,* and *IL-10* mRNA (**D**) was analyzed by RT-qPCR. (**E**) Monocytes were transfected with siRNA targeting AhR (siAhR) or negative control (siNC) for 1 day, followed by stimulation with IS for 24 hr. After a resting period of 5 days, cells were re-stimulated with LPS for 24 hr. mRNA expression levels of *TNF-α* and *IL-6* were assessed using RT-qPCR. (**F**) Enrichment of H3K4me3 on promoters of *TNFA* and *IL6* loci was assessed on day 6 after IS-training. 1% input was used as a normalization control. n=5 ~ 8. Bar graphs show the mean ± SEM. *=p < 0.05, **=p < 0.01, and ***=p < 0.001 by two-tailed paired *t*-test.

The online version of this article includes the following source data and figure supplement(s) for figure 4:

**Source data 1.** Raw data for *Figure 4A–F*.

**Source data 2.** PDF file containing *Figure 4A* and the relevant western blot analysis with highlighted bands and sample labels.

*Figure 4 continued on next page*

*Figure 4 continued*

**Source data 3.** Original image files for all western blot bands analyzed in *Figure 4A*.

**Figure supplement 1.** Indoxyl sulfate (IS)-mediated metabolic rewiring in IS-trained macrophages is independent of aryl hydrocarbon receptor (AhR).

**Figure supplement 1—source data 1.** Raw data for *Figure 4—figure supplement 1A–D*.

**Figure supplement 1—source data 2.** PDF file containing *Figure 4—figure supplement 1D* and the relevant western blot analysis with highlighted bands and sample labels.

**Figure supplement 1—source data 3.** Original image files for all western blot bands analyzed in *Figure 4—figure supplement 1D*.

Histone-modifying enzymes such as lysine demethylase (KDM) and lysine methyltransferase (KMT) are linked to the induction of trained immunity by remodeling the epigenetic status of cells (*Netea et al., 2020*; *Arts et al., 2016b*). However, RNA-Seq data of IS-trained macrophages showed no obvious change in the expression profile of major histone-modifying enzymes (*Figure 5—figure supplement 2A*). In agreement with this, mRNA expression of major histone modifying enzymes including *KDM5A*, *KDM5B*, *KDM5C*, *SETDB2*, *SETD7*, and *SETD3* were not changed in IS-trained macrophages on day 6 (*Figure 5—figure supplement 2B*; *Arts et al., 2016b*; *Kimball et al., 2019*; *Zhong et al., 2018*; *Keating et al., 2020*). Treatment with MTA, a non-selective methyltransferase inhibitor, partially inhibited expression of *ALOX5* and *ALOX5AP* mRNA (*Figure 5—figure supplement 2C*), suggesting limited epigenetic regulation of the AA pathway (*Figure 5—figure supplement 2C*). Ultimately, to validate the association between ChIP-Seq and RNA-Seq data, we employed Spearman's correlation for comparative analysis and conducted linear regression to ascertain the presence of a consistent global trend in RNA expression. Our findings unveiled a significant positive correlation, underscoring the consistent relationship between H3K4me3 enrichment and gene expression (*Figure 5—figure supplement 2D*).

## IS-induced trained immunity is validated by ex vivo and in vivo models

Since peripheral monocytes in ESRD patients are chronically exposed to uremic toxins like IS, we examined whether monocytes purified from ESRD patients before hemodialysis exhibit features of IS-trained macrophages. Ex vivo monocytes of ESRD patients had a higher production of TNF-α and mRNA expression of *IL-1β* and *MCP-1* after LPS stimulation than those of HCs (*Figure 6—figure supplement 1A–C* and *Table 2*). More importantly, monocytes of ESRD patients, which were trained by resting in the culture media for 6 days, significantly augmented production of TNF-α and IL-6 and expression of *IL-1β* mRNA upon LPS stimulation compared to those from HCs (*Figure 6A–C*). Consistent with our findings (*Figure 5*), the expression of ALOX5 in ex vivo monocytes of ESRD patients was significantly increased at the protein level compared with that of age-matched HCs (*Figure 6D–F*). Moreover, monocyte-derived macrophages (MDMs) from ESRD patients also had higher expression of ALOX5 compared to MDMs of HCs (*Figure 6G and H*), suggesting that IS in serum of ESRD patients contributes to the induction of trained immunity of monocytes/macrophages.

To examine the systemic in vivo effect of IS-trained immunity, we adopted a murine model in which IS was intraperitoneally injected daily for 5 days, followed by training for another 5 days and then re-stimulation with 5 mg/kg LPS for 75 min (*Figure 6I*). The level of TNF-α in serum was increased in IS-trained mice compared to that of control mice (*Figure 6J*). To further investigate the impact of IS-training on innate responses, splenic myeloid cells were isolated after 5 days of training (prior to injection with LPS) followed by in vitro stimulation with 10 ng/ml LPS for 24 hr. The amount of TNF-α and IL-6 in the supernatant was augmented following culture with LPS-stimulated mouse splenic myeloid cells derived from IS-trained mice compared the control condition (*Figure 6K*). Additionally, we observed upregulation of ALOX5 expression in ex vivo splenic myeloid cells of IS-treated mice compared to control mice (*Figure 6L and M*), which was similar to what was observed in monocytes and macrophages from ESRD patients (*Figure 6E–H*). Finally, treatment with zileuton, an ALOX5 inhibitor, inhibited the production of TNF-α and IL-6 in ex vivo splenic myeloid cells of IS-trained mice (*Figure 6N*).

Subsequently, we examined the impact of IS-trained immunity on mouse bone marrow-derived macrophages (BMDM). It was observed that BMDM from IS-trained mice did not exhibit heightened production of TNF-α and IL-6, and the expression level of ALOX5 in bone marrow progenitor cells remained unaltered when compared to non-trained cells, indicating that our acute IS-trained mice did

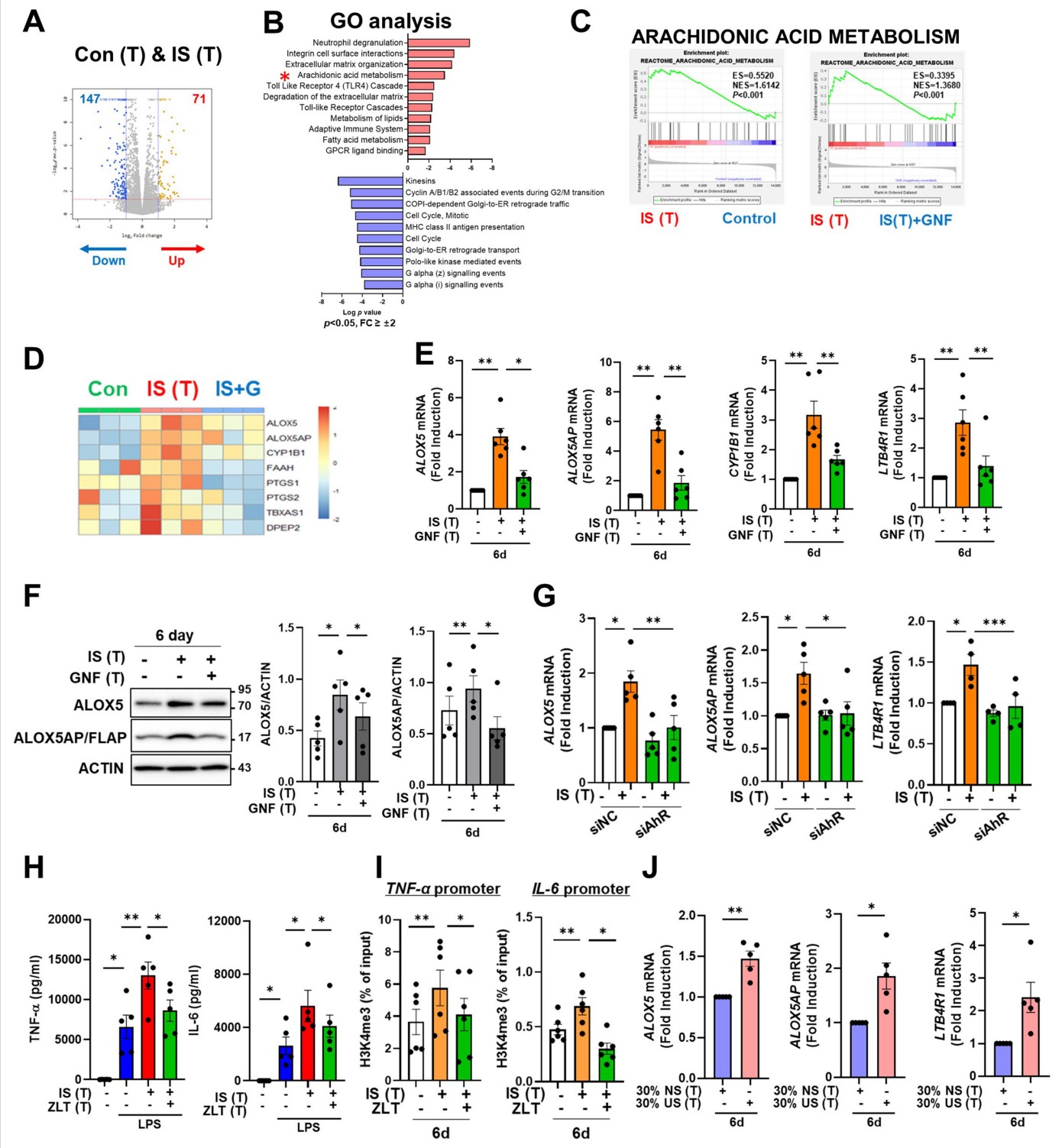

**Figure 5.** Aryl hydrocarbon receptor (AhR)-dependent induction of the arachidonic acid pathway contributes to indoxyl sulfate (IS)-induced trained immunity. (**A**) RNA-sequencing (RNA-Seq) analysis was performed on IS (1000 μM)-trained monocytes. Volcano plots show differentially expressed genes between IS-trained and non-trained macrophages. (**B**) Functional annotation of upregulated or downregulated genes (FC >±2, p<0.05) in IS-trained macrophages analyzed by Gene Ontology (GO) analysis with the Reactome Gene Set. (**C, D**) Gene Set Enrichment Analysis (GSEA) (**C**) and heatmap (**D**) of genes related to the AA metabolism in IS-trained macrophages compared to non-trained cells or compared to IS-trained macrophages with GNF351 (10 μM) treatment were analyzed. (**E, F**) On day 6 after IS-training with or without GNF351, expression of *CYP1B1*, arachidonate 5-lipoxygenase (*ALOX5*), ALOX5 activating protein (*ALOX5AP*), and *LTB4R1* mRNAs were quantitated using RT-qPCR (**E**) and cell lysates were prepared

*Figure 5 continued on next page*

*Figure 5 continued*

and immunoblotted for ALOX5 and ALOX5AP proteins (**F**). Band intensity in immunoblots was quantified by densitometry. β-ACTIN was used as a normalization control. (**G**) Monocytes were transfected with siRNA targeting AhR (siAhR) or negative control (siNC) for 1 day, followed by stimulation with IS for 24 hr. After a resting period of 5 days, mRNA expression level of each gene was assessed using RT-qPCR. (**H**) Monocytes were pretreated with zileuton (ALOX5 inhibitor, 100 μM) and trained with IS for 6 days followed by restimulation with lipopolysaccharide (LPS) (10 ng/ml) for 24 hr. TNF-α and IL-6 in supernatants were quantified by enzyme-linked immunosorbent assay (ELISA). (**I**) A chromatin immunoprecipitation (ChIP) assay was performed in IS-trained macrophages pre-treated with zileuton. 2% input was used as a normalization control. (**J**) The pooled normal serum (NS) from healthly controls (HCs) or uremic serum (US) from patients with end-stage renal disease (ESRD) were used to treat monocytes isolated from HCs for 24 hr at 30% (v/v) followed by resting for 5 days. Expression of *ALOX5*, *ALOX5AP*, and *LTB4R1* mRNAs were quantitated using RT-qPCR in trained macrophages with NS or US for 6 days. n=5 ~ 6. Bar graphs show the mean ± SEM. *=p < 0.05, **=p < 0.01, ***=p < 0.001 by two-tailed paired *t*-test.

The online version of this article includes the following source data and figure supplement(s) for figure 5:

**Source data 1.** Raw data for *Figure 5E–J*.

**Source data 2.** PDF file containing *Figure 5F* and the relevant western blot analysis with highlighted bands and sample labels.

**Source data 3.** Original image files for all western blot bands analyzed in *Figure 5F*.

**Figure supplement 1.** Aryl hydrocarbon receptor (AhR)-dependent induction of the arachidonic acid pathway contributes to indoxyl sulfate (IS)-induced trained immunity.

**Figure supplement 1—source data 1.** Raw data for *Figure 5—figure supplement 1C, D, F, and H–J*.

**Figure supplement 1—source data 2.** PDF file containing *Figure 5—figure supplement 1I* and the relevant western blot analysis with highlighted bands and sample labels.

**Figure supplement 1—source data 3.** Original image files for all western blot bands analyzed in *Figure 5—figure supplement 1I*.

**Figure supplement 2.** No obvious changes in expression of major histone-modifying enzymes were observed in indoxyl sulfate (IS)-induced trained immunity.

**Figure supplement 2—source data 1.** Raw data for *Figure 5—figure supplement 2B and C*.

**Table 2.** Demographic characteristics in study population.

|  | ESRD (N=21) | HCs (N=20) |
|---|---|---|
| **Clinical variables** | | |
| Age (years) | 62.4±12.4 | 56.9±7.8 |
| Male gender (%) | 15 (71.4%) | 8 (40%) |
| CAD (%) | 6 (28.6%) | |
| Hypertension (%) | 18 (85.7%) | |
| DM (%) | 7 (33.3%) | |
| SBP (mmHg) | 137.0±26.1 | |
| DBP (mmHg) | 64.5±20.4 | |
| Dialysis Duration (year) | 10.9±9.3 | |
| **Laboratory variables** | | |
| WBC count (X $10^3$ /μL) | 5.5±2.0 | |
| Hemoglobin (g/dL) | 11.2±1.7 | |
| Total cholesterol (mg/dL) | 152.6±37.3 | |
| BUN (mg/dL) | 51.5±19.9 | |
| Creatinine (mg/dL) | 8.5±3.8 | |
| Albumin (g/dL) | 3.9±0.5 | |
| Calcium (mg/dL) | 8.4±0.6 | |
| Phosphorus (mg/dL) | 4.7±1.6 | |
| hsCRP (mg/dL) | 4.9±8.6 | |

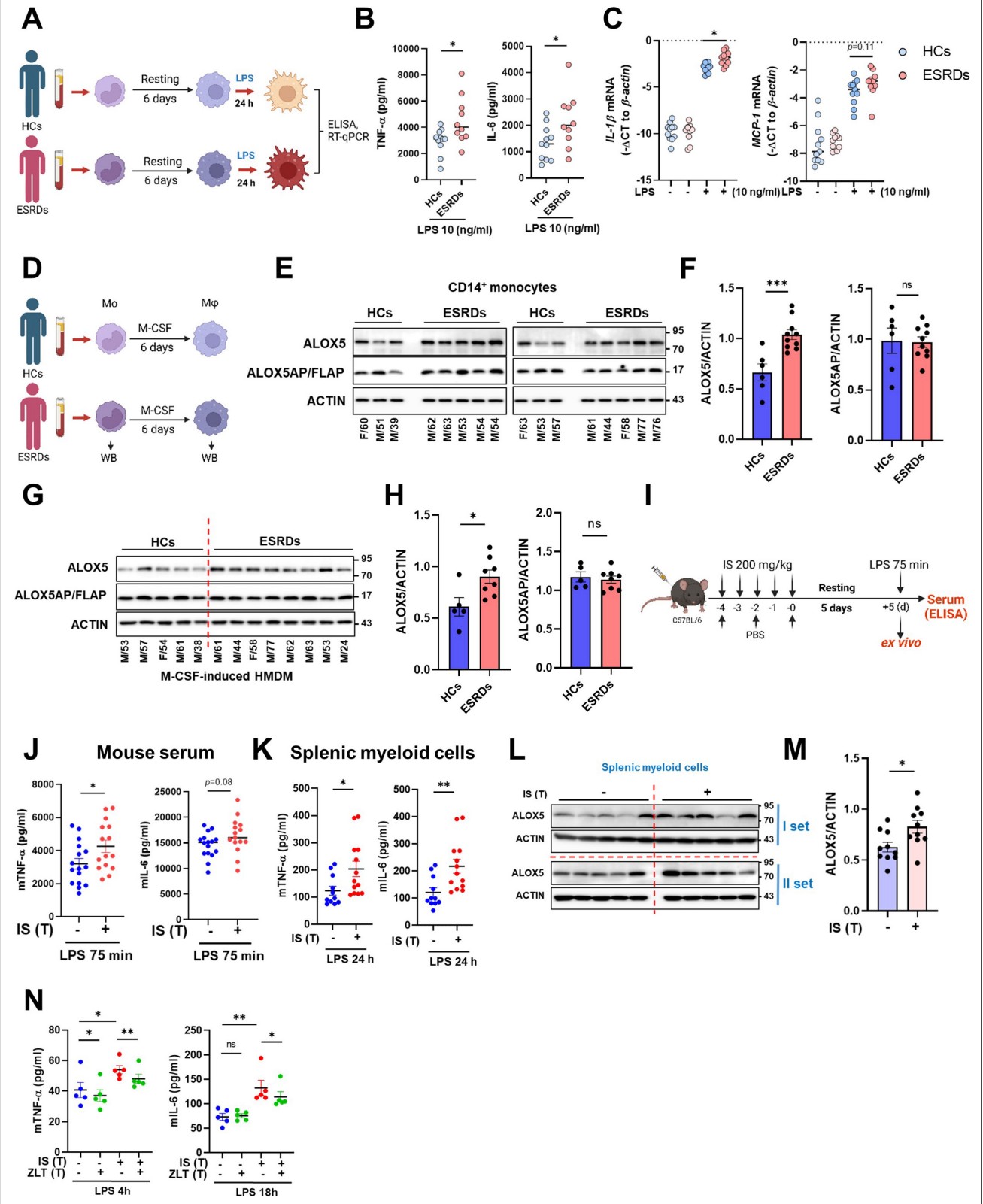

**Figure 6.** Ex vivo and in vivo validation of indoxyl sulfate (IS)-induced trained immunity. (**A–C**) CD14+ monocytes from end-stage renal disease (ESRD) patents (n=10) and age-matched healthy controls (HCs) (n=11) were rested for 6 days and stimulated by lipopolysaccharide (LPS) (10 ng/ml) for 24 hrs (**A**). TNF-α and IL-6 in supernatants were quantified by enzyme-linked immunosorbent assay (ELISA) (**B**) and mRNA expression of *IL-1β* and *MCP-1* were quantitated using RT-qPCR (**C**). (**D–G**) Arachidonate 5-lipoxygenase (ALOX5) and ALOX5 activating protein (ALOX5AP) protein levels in monocytes of (**E,**

*Figure 6 continued on next page*

*Figure 6 continued*

F) and in M-CSF-derived HMDM (**G, H**) of ESRD patients and HCs were analyzed by immunoblot analysis. Band intensity in immunoblots was quantified by densitometry. β-ACTIN was used as a normalization control. (**I**) C57BL/6 mice were injected daily with 200 mg/kg IS for 5 days and rested for another 5 days prior to LPS (5 mg/kg) treatment. Mice were sacrificed at 75 min post-LPS injection. (**J**) TNF-α and IL-6 in serum were quantified by ELISA (n=15 ~ 16). (**K**) Before LPS injection, IS-trained mice were sacrificed, and spleens were mechanically separated. Isolated splenic myeloid cells were treated ex vivo with LPS (10 ng/ml) for 24 hr and TNF-α and IL-6 in supernatants were quantified by ELISA (n=11 ~13). (**L, M**) The level of ALOX5 protein in splenic myeloid cells isolated from IS-trained or control mice was analyzed by western blot. The graph shows the band intensity quantified by the densitometry (**M**). (**N**) Isolated splenic myeloid cells were treated ex vivo with LPS (10 ng/ml), along with zileuton (100 μM). The levels of TNF-α and IL-6 in the supernatants were quantified using ELISA (n=5). The graphs show the median (**B–C**) or the mean ± SEM (**F–N**). *=p < 0.05, **=p < 0.01, and ***=p < 0.001 by unpaired non-parametric *t*-test or by two-tailed paired *t*-test between zileuton treatment group and no-treatment group (**N**).

The online version of this article includes the following source data and figure supplement(s) for figure 6:

**Source data 1.** Raw data for *Figure 6B, C, E–H and J–N*.

**Source data 2.** PDF file containing *Figure 6E, G and L* and the relevant western blot analysis with highlighted bands and sample labels.

**Source data 3.** Original image files for all western blot bands analyzed in *Figure 6E, G and L*.

**Figure supplement 1.** Ex vivo monocytes of end-stage renal disease (ESRD) patients exhibit features of IS-trained macrophages.

**Figure supplement 1—source data 1.** Raw data for *Figure 6—figure supplement 1B–E*.

**Figure supplement 1—source data 2.** PDF file containing *Figure 6—figure supplement 1E* and the relevant western blot analysis with highlighted bands and sample labels.

**Figure supplement 1—source data 3.** Original image files for all western blot bands analyzed in *Figure 6—figure supplement 1E*.

not induce central trained immunity (*Figure 6—figure supplement 1D and E*; *Kaufmann et al., 2018*; *Riksen et al., 2023*). Collectively, these findings offer compelling evidence for the involvement of IS and the induction of the AA pathway in the establishment of trained immunity in both monocytes and macrophages, both ex vivo and in vivo. Together, these data provide evidence for the role of IS and the induction of the AA pathway in the establishment of trained immunity of monocytes and macrophages both ex vivo and in vivo.

## Discussion

Recent studies have reported that in addition to pathogenic stimuli, endogenous sterile inflammatory insults including ox-LDL, hyperglycemia and uric acid, also trigger trained immunity and contribute to chronic inflammation in cardiovascular diseases and gout (*Christ et al., 2018*; *Edgar et al., 2021*; *Bekkering et al., 2014*; *Cabău et al., 2020*). In the present study, we provide evidence that IS, a major uremic toxin, provokes trained immunity in human monocytes/macrophage through epigenetic modification, metabolic rewiring, and AhR-dependent induction of the AA pathway, suggesting its important role in inflammatory immune responses in patients with CKD (*Figure 7*).

CKD is associated with increased risk factors of CVD including traditional risk factors, such as hypertension, age and dyslipidemia, as well as non-traditional risk factors, such as oxidative stress and inflammation (*Lim et al., 2021*; *Menon et al., 2005*). Further, recent cohort studies have shown that CKD is an independent risk factor for CVD (*Menon et al., 2005*). Uremia accompanying renal failure causes immune dysfunction, which is closely linked to the pathogenesis of CKD-related CVD (*Betjes, 2013*). Among over 100 uremic toxins identified, IS is a prototypical protein-bound uremic toxin most likely to be participating in progressive pathophysiology of CVD including endothelial dysfunction, vascular calcification, and increased atherosclerosis (*Kim et al., 2017*; *Nakano et al., 2019*; *Hung et al., 2017*; *Leong and Sirich, 2016*). Mounting evidence suggests that a prolonged hyperactivation of trained immunity is intimately related to the pathogenesis of atherosclerosis, the major contributor to cardiovascular diseases. Oxidized low-density lipoprotein (oxLDL), hyperglycemia, and the estern diet, all known to be associate with the progression of atherosclerosis, have been reported to induce trained immunity through epigenetic reprogramming (*Christ et al., 2018*; *Edgar et al., 2021*; *Bekkering et al., 2014*). These findings suggest that IS plays a role as a typical endogenous inflammatory insult in activating monocytes and macrophages and modulating their responses. Given that

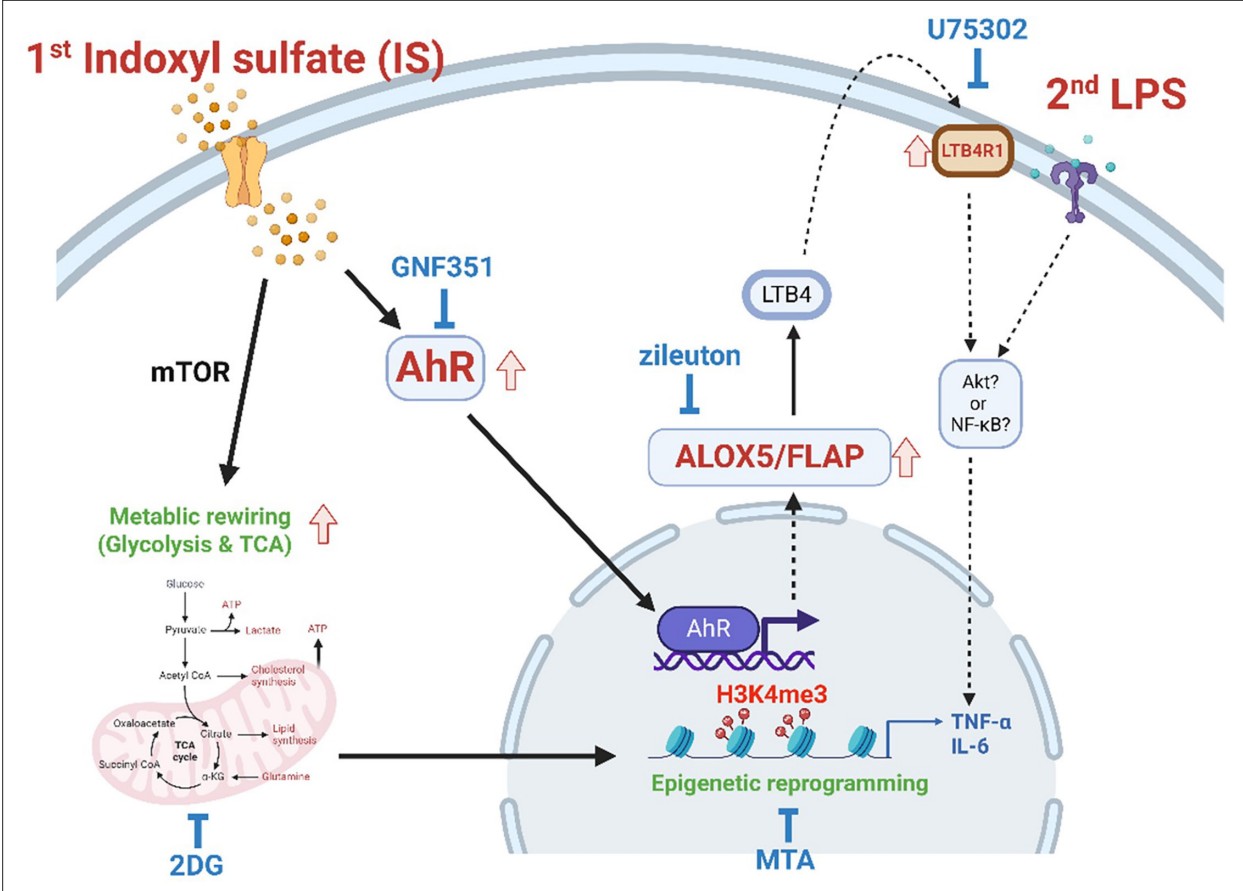

**Figure 7.** Proposed mechanism of indoxyl sulfate (IS)-induced trained immunity. IS-induced trained immunity in human monocytes is mediated by epigenetic reprogramming and metabolic rewiring via histone modification H3K4m3 and increased glycolysis and mitochondrial respiration, respectively. Direct interaction of uremic toxin IS with the aryl hydrocarbon receptor (AhR) in human monocytes activates AhR signaling pathways that are involved in enhanced expression of the arachidonic acid metabolism-related genes arachidonate 5-lipoxygenase (ALOX5), ALOX5 activating protein (ALOX5AP), and LTB4R1 and augmented production of TNF-α and IL-6 upon stimulation with lipopolysaccharide (LPS) as secondary stimulus via epigenetic regulation. A pivotal role of each pathway or molecule was confirmed by in vitro assay with inhibitors including GNF351 (an AhR antagonist), zileuton (an ALOX5 inhibitor), U75302 (a BLT1 receptor inhibitor), 2-deoxy-d-glucose (2-DG) (a glycolysis inhibitor), and 5'-methylthioadenosine (MTA) (a non-selective methyltransferase inhibitor). Meanwhile, the AhR-independent mechanism contributes to metabolic rewiring, such as increased glycolysis in IS-trained macrophages, which leads to enhanced proinflammatory responses upon secondary stimulation.

IS is difficult to clear by hemodialysis, this toxin has a chronic effect on the immune system of patients. Nonetheless, little is known about the effects of IS on trained immunity.

As observed using a common in vitro model of trained immunity established by Netea and other groups (*Figure 1A*), CD14+ monocytes, which are exposed for 24 hr to the first insult with IS and rested for 5 days without IS, produced an augmented level of TNF-α and IL-6 and decreased level of IL-10 in response to an unrelated second insult with LPS or Pam3cys, which is a feature typical of trained immunity of monocytes (*Figure 1B–E*). In contrast to TNF-α and IL-6, major proinflammatory cytokines, IL-10 exerts potent deactivating effects on macrophages and T cells, influencing various cellular processes in inflammatory diseases (*Mallat et al., 1999*; *Wei et al., 2022*). Additionally, it is noteworthy that IL-10-deficient macrophages exhibit an augmentation in the proinflammatory cytokine TNF-α (*Smallie et al., 2010*; *Couper et al., 2008*). Therefore, the reduced gene expression of IL-10 by IS-trained monocytes may contribute to the heightened expression of proinflammatory cytokines. Mechanistic studies have demonstrated that the induction of trained immunity is coordinated through the interplay of epigenetic modifications and metabolic rewiring, which is broadly characterized as prolonged changes in transcription programs and cell physiology that do not involve

permanent genetic changes, such as the mutations and recombination events crucial for adaptive immunity (*Netea et al., 2016*). In the present study, ChIP-Seq and real-time metabolic analysis show that the induction of IS-trained immunity in human monocytes is attributable to epigenetic modification and metabolic rewiring (*Figures 2 and 3*). Consistent with previous findings (*Bekkering et al., 2018*; *Bekkering et al., 2014*), trimethylation of histones at H3K4 on *TNFA* and *IL6* promoters was increased in IS-trained macrophages and maintained even after secondary stimulation with LPS (*Figure 3B* and *Figure 3—figure supplement 1B*). Furthermore, the production of TNF-α and IL-6 upon LPS stimulation was completely inhibited by pretreatment with MTA, a methyltransferase inhibitor (*Figure 3C*), demonstrating that IS-induced trained immunity is associated with epigenetic modification. Moreover, H3K4me3-ChIP-Seq data showed that IS-induced trained immunity accompanied by epigenetic reprogramming and H3K4me3 was enriched in genes related to activation of the innate immune responses as illustrated by GO analysis (*Figure 3E–G*).

We identified AhR as a critical mediator of IS-trained immunity in human monocytes (*Figure 4*). AhR is a ligand-activated nuclear transcription factor (TF), which is activated by several exogenous compounds, such as benzo[a]pyrene environmental pollutants and 2,3,7,8-tetrachloro-dibenzo-*p*-dioxin (TCDD), as well as by multiple endogenous ligands including tryptophan and indole metabolites (*Rothhammer and Quintana, 2019*). AhR plays a multifaceted role in modulating cellular mechanisms such as inflammation, cell growth, and antioxidant responses (*Stockinger et al., 2014*). AhR is expressed by various immune cells, and its signaling exerts integrative effects on the cellular environment and metabolism of the immune responses (*Gutiérrez-Vázquez and Quintana, 2018*). However, little is known about the role of AhR in the induction of trained immunity.

In the present study, we show that IS-trained immune responses, characterized by the expression of proinflammatory cytokines and chemokines, were attenuated by GNF351, an AhR antagonist, or through the knockdown of AhR using siRNA (*Figure 4B–E*). This inhibition was accompanied by repression of enriched H3K4me3 on *TNFA* and *IL6* promoters in IS-trained macrophages (*Figure 4F*), indicating an AhR-dependent mechanism. However, increased glycolysis and mitochondria respiration in IS-trained macrophages were not suppressed by the blockade of AhR activation with GNF351, suggesting the AhR activation is not directly involved in metabolic rewiring in IS-trained immunity (*Figure 4—figure supplement 1B–D*).

Metabolic rewiring, especially upregulation of aerobic glycolysis, is known as a major mechanism underlying the induction of trained immunity via regulation of epigenetic modification by metabolites, such as mevalonate and fumarate generated from this metabolic rewiring (*Bekkering et al., 2021*). Furthermore, in addition to establishing the bidirectional link between heightened glycolysis and activating histone modifications, as evidenced by the suppression of histone modification by 2DG, a glycolysis inhibitor, in IS-induced immunity (*Figure 3D*), recent studies have demonstrated that immune priming long non-coding RNAs (IPLs) also induce epigenetic reprogramming by influencing 3D nuclear architecture (*Netea et al., 2020*; *Fanucchi et al., 2021*) and that RUNX1, a transcription factor, contributes to the induction of trained immunity by overexpression of RUNX1 target genes (*Edgar et al., 2021*), suggesting that a variety of mechanism is involved in epigenetic modification in trained immunity.

Although inducers of trained immunity, such as β-glucan, BCG, uric acid, and oxLDL, initiate intracellular signaling and metabolic pathways, each via different receptors, the most common pathway is the Akt/mTOR/HIF1α-dependent induction of aerobic glycolysis (*Netea et al., 2020*; *Mulder et al., 2019*; *Bekkering et al., 2021*). Our data also revealed that training with IS led to enhanced glycolysis, which is critical for the production of TNF-α and IL-6 upon LPS stimulation as confirmed by experiments with 2DG, a glycolysis inhibitor (*Figure 2E*). Recent studies have shown that IS activates mTORC1 in a variety of cells such as epithelial cells, fibroblasts, and THP-1 cells mainly via the organic anion transporters (OAT)/NADPH oxidase/ROS pathway, but not the AhR pathway (*Nakano et al., 2021*). Our findings suggest that IS-trained macrophages acquire the characteristics of trained immunity by AhR-dependent and -independent mechanisms and enhances proinflammatory responses upon secondary stimulation. Thus, addressing the mechanism underlying AhR-independent metabolic rewiring in IS-trained macrophages will require further investigations.

A finding of particular interest in our study is that the induction of IS-induced trained immunity is dependent on the AhR-ALOX5/ALOX5AP axis, as depicted by RNA-Seq analysis and confirmatory in vitro analysis of mRNA and protein expression (*Figure 5* and *Figure 5—figure supplement 1*).

ALOX5 and ALOX5AP are major rate-limiting enzymes associated with the AA pathway, involved in the production of leukotrienes, proinflammatory lipid mediators derived from AA (*Rådmark et al., 2007*). Among leukotrienes, leukotriene B4 (LTB4), an extremely potent inflammatory mediator, binds to Gprotein-coupled protein, LTB4R, and enhances inflammatory responses by increased phagocytosis and activation of the signaling pathway for the production of cytokines (*Salina et al., 2020*; *Pernet et al., 2019*). Mechanistically, the LTB4-LTB4R signaling pathway induces the PI3K/Akt or NF-κB pathways (*Ihara et al., 2007*; *Tong et al., 2005*). AhR-dependent LTB4 production through enhanced ALOX5 expression in hepatocytes reportedly induces hepatotoxicity via neutrophil infiltration (*Takeda et al., 2017*). The increase in ALOX5 activity and LTB4 expression has been reported in patients with ESRD (*Maccarrone et al., 2002*; *Maccarrone et al., 2001*; *Montford et al., 2019*) and ALOX5 mediates mitochondrial damage and apoptosis in mononuclear cells of ESRD patients (*Maccarrone et al., 2002*). Thus, antagonists of ALOX5/ALOX5AP have been used for treatment in CKD (*Montford et al., 2019*). Consistent with these findings, peripheral monocytes derived from ESRD patients in our cohort have increased expression of ALOX5 and this increase was maintained after differentiation into macrophages with M-CSF (*Figure 6E–H*). Considering that pretreatment with the uremic serum of ESRD patients elicits a change in gene expression and cytokine production as observed in IS-trained macrophages, it is likely that increased ALOX5 in monocytes and macrophages of ESRD patients is mediated by IS in uremic serum. Our previous studies have shown that IS is an important uremic toxin in the serum of ESRD patients that elicits proinflammatory responses of monocytes (*Kim et al., 2017*). Furthermore, the AhR-ALOX5-LTB4R1 pathway is involved in IS-induced trained immunity of patients with ESRD.

Recent studies have demonstrated that trained immunity of macrophages is implicated in the pathogenesis of CVD, such as atherosclerosis (*Edgar et al., 2021*; *Bekkering et al., 2014*; *van der Valk et al., 2016*; *Mulder et al., 2019*). Considering a higher plasma level of IS in ESRD patients even after hemodialysis (*Duranton et al., 2012*; *Lim et al., 2021*), this is potentially difficult to reconcile with trained immunity in which there typically is a short exposure to the training stimulus followed by a period of rest (*Netea et al., 2016*; *Netea et al., 2020*). However, when ESRD monocytes exposed to the IS in the circulation enter atherosclerotic plaques and differentiate into macrophages, they are no longer exposed to IS, because this is protein-bound and hence not expected to the present in the plaque microenvironment.

Murine models have been widely used to investigate the specific contribution of inducers of trained immunity and the underlying mechanisms to a long-term functional modification of innate cells in vivo (*Saeed et al., 2014*; *Keating et al., 2020*; *Garcia-Valtanen et al., 2017*). Mice trained with β-glucan enhance the production of proinflammatory cytokines such as TNF-α, IL-6, and IL-1β by monocytes and macrophages in response to secondary microbial stimuli and subsequently obtain increased protection against various microbial infections (*Mulder et al., 2019*; *Bekkering et al., 2021*). Our in vivo and ex vivo mouse experiments for trained immunity demonstrate that IS-trained immunity has biological relevance (*Figure 6I*). Previous studies have shown that intraperitoneal injection of exogenous IS into wild-type C57BL/6 mice leads to an increase of IS in the plasma until 3~6 hr post-injection despite its rapid excretion by the kidney. In this period, the expression of pro-inflammatory, pro-oxidant, and pro-apoptotic genes in peritoneal macrophages was upregulated (*Nakano et al., 2021*; *Rapa et al., 2021*). Moreover, intraperitoneal injection of IS daily for 3 days elevates IS in the plasma of mice and activates the mTORC1 signaling pathway in the mouse kidney (*Nakano et al., 2021*). This suggests that the IS-injected mouse model is suitable for investigating the mechanisms underlying IS-mediated immune responses. Our data show increased TNF-α in serum after injection of LPS (*Figure 6J*). Moreover, ALOX5 protein expression was increased in splenic myeloid cells derived from IS-trained mice and ex vivo stimulation with LPS for 24 hr induced TNF-α and IL-6 expression in these splenic myeloid cells (*Figure 6K–M*). Thus, this suggests a systemic induction of IS-trained immunity in the mouse model.

In conclusion, the current study provides new insight into the role of IS as an inducer of trained immunity as well as the underlying mechanisms in human monocytes/macrophages by investigating the effect of IS in vivo and in vitro using experimental models of trained immunity. Here, we demonstrate that IS, a major uremic toxin, induces trained immunity characterized by the increased proinflammatory TNF-α and IL-6 in human monocytes following secondary stimulation through epigenetic modification and metabolic rewiring. IS-mediated activation of AhR is involved in the induction of

trained immunity through enhanced expression of AA metabolism-related genes such as ALOX5 and ALOX5AP. Monocytes from patients with ESRD exhibit increased expression of ALOX5 and after 6 day resting, they exhibit enhanced TNF-α and IL-6 production to LPS. Furthermore, the uremic serum of ESDR patients causes HC-derived monocytes to increase the production of TNF-α and IL-6 upon LPS re-stimulation, implying IS-mediates trained immunity in patients. Supporting our in vitro findings, mice trained with IS and their splenic myeloid cells had increased production of TNF-α after in vivo and ex vivo LPS stimulation. These results suggest that IS plays an important role in the induction of trained immunity, which is critical in inflammatory immune responses in patients with CKD and thus, it holds potential as a therapeutic target.

## Materials and methods

### Key resources table

| Reagent type (species) or resource | Designation | Source or reference | Identifiers | Additional information |
|---|---|---|---|---|
| Transfected construct (human) | ON-TARGETplus Human AHR siRNA | Dharmacon | L-004990-00-0005 | Transfected construct (human) |
| Transfected construct (human) | ON-TARGETplus Human ALOX5 siRNA | Dharmacon | L-004530-00-0005 | Transfected construct (human) |
| Biological sample (human) | Primary human CD14$^+$ monocytes | Blood from healthy donors or ESRD patients. | The institutional review board of Seoul National University Hospital and Severance Hospital | Freshly isolated from blood of donors |
| Antibody | Anti-AhR (D5S6H) antibody (rabbit monoclonal) | Cell Signaling Technology | #83200 | WB (1:1000) |
| Antibody | 5-Lipoxygenase (C49G1) antibody (rabbit monoclonal) | Cell Signaling Technology | #3289 | WB (1:1000) |
| Antibody | Recombinant Anti-FLAP antibody [EPR5640] (rabbit monoclonal) | Abcam | ab124714 | WB (1:1000) |
| Antibody | Tri-Methyl-Histone H3 (Lys4) (C42D8) antibody (rabbit mAb) | Cell Signaling Technology | #9751 | ChIP (3–5 µl per sample) |
| Sequence-based reagent | Primer for RT-qPCR | This paper | | *Table 3* in this paper |
| Sequence-based reagent | Primer for ChIP assay | *Bekkering et al., 2018*; *Arts et al., 2016a* | | *Table 4* in this paper |
| Commercial assay or kit | TNF alpha Human Uncoated ELISA Kit | Invitrogen | 88-7346-86 | |
| Commercial assay or kit | IL-6 Human Uncoated ELISA Kit | Invitrogen | 88-7066-88 | |
| Commercial assay or kit | TNF alpha Mouse Uncoated ELISA Kit | Invitrogen | 88-7324-88 | |
| Commercial assay or kit | IL-6 Mouse Uncoated ELISA Kit | Invitrogen | 88-7064-88 | |
| Chemical compound, drug | Indoxyl sulfate potassium salt | Sigma-Aldrich | I3875 | |
| Chemical compound, drug | GNF351 | Sigma-Aldrich | 182707 | |
| Chemical compound, drug | LPS from *E. coli* O111:B4 for in vitro experiments | Invivogen | tlrl-eblps | |
| Chemical compound, drug | Zileuton | Sigma-Aldrich | Z4277 | |
| Chemical compound, drug | 5'-Deoxy-5'-(methylthio)adenosine (MTA) | Sigma-Aldrich | D5011 | |

*Continued on next page*

*Continued*

| Reagent type (species) or resource | Designation | Source or reference | Identifiers | Additional information |
|---|---|---|---|---|
| Chemical compound, drug | 2-Deoxy-D-glucose | Sigma-Aldrich | D6134 | |
| Chemical compound, drug | Human serum | Sigma-Aldrich | H6914 | |
| Software, algorithm | Graph Pad Prism 8 | Graphpad software | https://www.graphpad.com/ | |
| Software, algorithm | Image J | NIH | https://imagej.nih.gov/ij/download.html | |
| Software, algorithm | Biorender | Biorender | https://app.biorender.com/user/signin | |
| Other | Raw data files for ChIP-seq | This paper | GSE263019 | https://www.ncbi.nlm.nih.gov/geo/query/acc.cgi?acc=GSE263019 |
| Other | Raw data files for RNA-seq | This paper | GSE263024 | https://www.ncbi.nlm.nih.gov/geo/query/acc.cgi?acc=GSE263024 |

## Human monocyte preparation and culture

Peripheral blood mononuclear cells (PBMCs) were isolated from peripheral blood by density gradient centrifugation (Bicoll-Separating Solution; BIOCHROM Inc, Cambridge, UK). Monocytes were positively purified from PBMCs with anti-CD14 magnetic beads (Miltenyi Biotec Inc, Auburn, CA). For in vitro trained immunity experiments, purified monocytes were treated with IS for 24 h, followed by washing with pre-warmed PBS and incubation for another 5 days in RPMI medium supplemented with 10% human AB serum (HS, Sigma-Aldrich, St. Louis, MO), 100 U/ml penicillin, and 100 µg/ml streptomycin (Gibco, Grand Island, NY). On day 6, cells were re-stimulated with LPS or Pam3cys for 24 hr, and the supernatants and their lysates were collected and stored at –80°C until use. In some experiments, chemical inhibitors were used for a 1 hr pre-treatment at the indicated concentrations prior to the treatment with IS. To test the effect of uremic serum on the induction of trained immunity, CD14+ monocytes purified from HC donors were seeded into 48-well plates and incubated for 24 hr at 30% (v/v) with the pooled uremic sera (US) from ESRD patients or the pooled normal sera (NS) from HCs, followed by washing with pre-warmed PBS and incubation for another 5 days. On day 6, cells were re-stimulated with LPS for 24 hr. To examine whether monocytes from ESRD patients inhibit features of IS-trained immunity, CD14+ monocytes were purified from ESRD patients and age-matched HCs, followed by stimulation with LPS for 24 hr. In addition, purified monocytes were seeded and rested for 6 days in RPMI medium supplemented with 10% human AB serum, 100 U/ml penicillin, and 100 µg/ml streptomycin to induce trained immunity. On day 6, the cells were stimulated with LPS for 24 hr. In some experiments, MDMs were differentiated from purified CD14+ monocytes from ESRD patients or age-matched HCs in RPMI 1640 medium supplemented with 10% fetal bovine serum (FBS, BioWest, Nuaill´e, France), 50 ng/ml recombinant human M-CSF (PeproTech, Rocky Hill, NJ, USA), 100 U/ml penicillin, and 100 mg/ml streptomycin. On day 6, MDMs were used for immunoblot analysis.

## Chemicals and antibodies

IS potassium salt, GNF351, MTA, 2-DG, and zileuton were purchased from Sigma-Aldrich (Burlington, MA, USA). LPS from *E. coli* 0111: B4 were purchased from InvivoGen (San Diego, CA, USA) for in vitro experiments and Sigma-Aldrich for in vivo experiments. U-75302 was obtained from Cayman Chemical (Ann Arbor, Michigan, USA). Anti-AhR and anti-5-loxygenase (5-LOX) antibodies (Ab) for immunoblot assay and anti-trimethyl H3K4 (H3K4me3) Ab for ChIP were purchased from Cell Signaling Technology (Danvers, MA, USA). Anti-ALOX5AP (FLAP) antibody was obtained from Abcam Inc (Cambridge, UK).

## Enzyme-linked immunosorbent assay

The amounts of TNF-α and IL-6 in culture supernatants of LPS or Pam3cys-re-stimulated IS-trained macrophages were quantified using commercial human enzyme-linked immunosorbent assay (ELISA)

**Table 3.** Primers for qPCR.

| Gene name | Primer sequence (5'–3') |
|---|---|
| Human Actin | Forward: GGACTTCGAGCAAGAGATGG |
| | Reverse: AGCACTGTGTTGGCGTACAG |
| Human TNF-α | Forward: TGCTTGTTCCTCAGCCTCTT |
| | Reverse: CAGAGGGCTGATTAGAGAGAGGT |
| Human IL-6 | Forward: TACCCCCAGGAGAAGATTCC |
| | Reverse: TTTTCTGCCAGTGCCTCTTT |
| Human pro-IL-1β | Forward: CACGATGCACCTGTACGATCA |
| | Reverse: GTTGCTCCATATCCTGTCCCT |
| Human IL-10 | Forward: TGCCTTCAGCAGAGTGAAGA |
| | Reverse: GGTCTTGGTTCTCAGCTTGG |
| Human MCP-1 | Forward: AGCAGCAAGTGTCCCAAAGA |
| | Reverse: GGTGGTCCATGGAATCCTGA |
| Human ALOX5 | Forward: TCTTGGCAGTCACATCTCTTC |
| | Reverse: GAATGGGTCCCTATGGTGTTTA |
| Human ALOX5AP | Forward: GTCGGTTACCTAGGAGAGAGAA |
| | Reverse: GACATGAGGAACAGGAAGAGTATG |
| Human LTB4R1 | Forward: GTTCATCTCTCTGCTGGCTATC |
| | Reverse: AGCGCTTCTGCATCCTTT |
| Human CYP1B1 | Forward: TGCCTGTCACTATTCCTCATGCCA |
| | Reverse: ATCAAAGTTCTCCGGGTTAGGCCA |
| Human KDM5A | Forward: CAGCTGTGTTCCTCTTCCTAAA |
| | Reverse: CCTTCGAGACCGCATACAAA |
| Human KDM5B | Forward: GCCCTCAGACACATCCTATTC |
| | Reverse: AGTCCACCTCATCTCCTTCT |
| Human KDM5C | Forward: ACAGAAGGAGAAGGAGGGTAT |
| | Reverse: CACACACAGATAGAGGTTGTAGAG |
| Human SETDB2 | Forward: CCACTGAACTTGAAGGGAGAAA |
| | Reverse: GTGGAGTGCTGAAGAATGAGAG |
| Human SETD3 | Forward: TGGTTACAACCTGGAAGATGAC |
| | Reverse: CGTTGGATCGAGTGCCATAA |
| Human SETD7 | Forward: AGTGTAAACTCCCTGGCCCT |
| | Reverse: GTTCACGGAGAAAAGAACGG |

kits (Thermo Fisher Scientific, Waltham, MA, USA). Optical density was measured using the Infinite M200 (Tecan, Männedorf, Switzerland).

## Quantitative RT-PCR

Total RNA was prepared using RNA purification kit (Macherey-Nagel GmbH & Co. KG, Germany), followed by cDNA synthesis (Bio-line, London, UK), and then real-time quantitative RT-PCR was performed with the CFX system (Bio-Rad, Hercules, CA) using the SensiFAST SYBR Lo-ROX (Bio-line, London, UK). Sequences of primers used in this investigation are shown in *Table 3*. Normalization of

**Table 4.** Primers for ChIP-qPCR.

| Gene name | Primer sequence (5'–3') |
| --- | --- |
| Human TNF-α promoter | Forward: GTGCTTGTTCCTCAGCCTCT |
| | Reverse: ATCACTCCAAAGTGCAGCAG |
| Human IL-6 promoter | Forward: AGGGAGAGCCAGAACACAGA |
| | Reverse: GAGTTTCCTCTGACTCCATCG |
| Human HK2 promoter | Forward: GAGCTCAATTCTGTGTGGAGT |
| | Reverse: ACTTCTTGAGAACTATGTACCCTT |
| Human PFKP promoter | Forward: CGAAGGCGATGGGGTGAC |
| | Reverse: CATCGCTTCGCCACCTTTC |

gene expression levels against the expression of ACTINB using the comparative CT method ($\Delta\Delta$CT) was used for quantification of gene expression.

## ChIP-qPCR and ChIP-Seq

Cells were washed with Dulbecco's PBS and crosslinked for 5 min with 1% formaldehyde at room temperature (RT), followed by quenching with 100 mM glycine for 5 min. Cells were harvested with lysis buffer (50 mM HEPES, pH7.5, 140 mM NaCl, 1 mM EDTA, 10% glycerol, 0.5% NP-40 and 0.25% Triton X-100) with protease inhibitors on ice for 10 min and were then washed with washing buffer (10 mM Tris-HCl, pH7.0, 200 mM NaCl, 1 mM EDTA, and 0.5 mM EGTA) for 10 min. The lysates were resuspended and sonicated in sonication buffer (10 mM Tris-HCl, pH8.0, 100 mM NaCl, 1 mM EDTA, 0.5 mM EGTA, 0.1% sodium deoxylcholated and 0.5% N-laurolsarcosine) using a Bioruptor (diagenode, Denville, NJ) with 30 s on and 30 s off on a high-power output for 25 cycles. After sonication, samples were centrifuged at 12,000 rpm for 10 min at 4°C and 1% sonicated cell extracts were saved as input. Cell extracts were incubated with protein A agarose loaded with the H3K4me3 Ab overnight at 4°C, and then Ab-bound agarose beads were washed twice with sonication buffer, once with sonication buffer with 500 mM NaCl, once with LiCl wash buffer (10 mM Tris-HCl, pH8.0. 1 mM EDTA, 250 mM LiCl and 1% NP-40), and once with TE with 50 mM NaCl. After washing, DNA was eluted in freshly prepared elution buffer (1% SDS and 0.1 M NaHCO$_3$). Cross-links were reversed by overnight incubation at 65°C with RNase A, followed by incubation with proteinase K for 1 hr at 60°C. DNA was purified with NucleoSpin gDNA Clean-up Kit (Macherey-Nagel GmbH & Co. KG, Germany). For ChIP-qPCR assays, immunoprecipitated DNA was analyzed by quantitative real-time PCR and results were normalized against input DNA. The sequences of primers used for ChIP-qPCR are shown in *Table 4*; *Bekkering et al., 2018*; *Arts et al., 2016a*.

For ChIP-seq experiments, purified DNA were prepared for DNA libraries using TruSeq DNA Sample Prep Kit according to Library Protocol TruSeq ChIP Sample Preparation Guide 15023092 Rev. B. Next, illumina sequencing were performed using NovaSeq 6000 S4 Reagent Kit according to sequencing protocol of NovaSeq 6000 System User Guide Document # 1000000019358 v02. Sequenced reads were trimmed using Trimmomatic software. Fragments were aligned to hg19 using Bowtie2 software. Aligned fragments of H3K4me3-ChIP samples were concatenated into a single file to generate consistent peak ranges between samples using the makeTagDirectory function of Homer Suite. For each sample, regions of H3K4me3 enrichment compared to the input sample were collected using callpeaks function in MACS3 software. H3K4me3-rich regions from the same group of different donors were compared to peaks in linked samples using the findoverlap function of the GenomicRange R-package, and 11,123 peaks were collected for further analysis. For quantitative comparisons between IC-trained groups and controls, the number of fragments of each peak in BEDPE was collected using the coverage function of the BEDtools software. Then, the number of fragments in the peak was normalized to CPM and significance was compared using edgeR R-package. Finally, we selected 7,136 peaks with at least 15 CPM from the larger average group to exclude lowly H3K4me3 enriched peaks.

Enriched peaks were selected base on a p-value of 0.05 or less and log2 fold change of >1.3. The selected enriched peaks were used for Go pathway analysis. Pathway analysis was conducted using Metascape web-based platform (*Zhou et al., 2019*) and significant pathways were identified on the basis of Go biological process and Reactome gene sets. Significant pathways were selected with p<0.05 and enrichment score (ES) >1.5.

## RNA-Seq and analysis

After RNA extraction, libraries for sequencing were prepared using the TruSeq Stranded mRNA LT Sample Prep Kit and sequencing were performed using NovaSeq 6000 System User Guide Document # 1000000019358 (Illumina). To analyze RNA-Seq data, trimmed reads were aligned to the human GRCh37 (NCBI_105.20190906). Gene expression profiling was performed using StringTie and then read count and FPKM (Fragment per Kilobase of transcript per Million mapped reads) were acquired. DEGs were selected based on *p*-value of 0.05 or less. Selected data were applied to hierarchical cluster analysis to display basal and luminal differences and were further filtered according to gene expression levels with a log2 fold change of < –2 and >2. DEGs were visualized using the R (ver. 4.1.1) and pheatmap package (ver. 1.0.8). For GSEA, samples were categorized into distinct groups based on treatment conditions. All transcripts within annotated genes (~14,404 features in total) regarding expression values were uploaded to locally installed GSEA software (ver. 4.2.3) (*Subramanian et al., 2005*). The analysis was performed by comparing the 'IS (T)' group against the 'Control' group to identify differentially enriched gene sets within the Reactome pathway database, particularly focusing on the aAA metabolism pathway. A similar comparison was made between the 'IS (T)' and 'IS (T)+GNF' groups to assess the effect of the GNF inhibitor. Outputs were filtered based on a nominal p<0.05 and a normalized enrichment score (NES) >1.3 to determine statistical significance. These thresholds were applied to ensure the robustness of the findings in the context of multiple hypothesis testing. The enrichment plots were generated to visualize the distribution of the gene sets and their ESs. Lastly, pathway analyses were further substantiated using the Metascape web-based platform (*Zhou et al., 2019*), with significant pathways identified using DEGs and selected based on a p<0.05 and an ES >1.5.

## Metabolic analysis

To profile the metabolic state of the cells, CD14$^+$ monocytes were seeded onto XFe24 cell culture plates (Seahorse Bioscience, Lexington, MA) with RPMI medium with 10% HS, followed by the induction of trained immunity for 6 days as described in *Figure 1A*. Metabolic analysis on IS-trained macrophages was performed according to the manufacturer's instructions. For the glycolysis stress test, culture media was replaced with Seahorse XF Base media supplemented with 2 mM L-glutamine (pH7.4) and incubated for 1 hr in the non-CO2 incubator. Glucose (10 mM), oligomycin (2 μM), and 2-DG (50 mM, all from Sigma-Aldrich) were sequentially used to treat cells during real-time measurements of ECAR using Seahorse XFe24 Analyzer (Seahorse Bioscience). For the mito stress test, cells were incubated with Seahorse XF Base media supplemented with 1 mM pyruvate, 2 mM L-glutamine, and 10 mM glucose (pH7.4) for 1 hr in the non-CO2 incubator. Oligomycin (1.5 μM), FCCP (2 μM), and rotenone/antimycin A (0.5 μM, all from Sigma-Aldrich) were sequentially used to treat cells during real-time measurements of OCR using the Seahorse XFe24 Analyzer. Parameters of glycolysis stress test and mito stress test were calculated using Seahorse XF glycolysis or the mito stress test report generator program that was provided by the manufacturer.

## Immunoblot analysis

Total proteins were prepared using radioimmunoprecipitation assay (RIPA) buffer (150 mM NaCl, 10 mM Na$_2$HPO$_4$, 0.5% sodium deoxycholate, 1% NP-40) containing a protease and phosphatase inhibitor cocktail (Thermo Fisher Scientific, Waltham, MA, USA). Cell lysates were separated on an 8–12% SDS-polyacrylamide gel and blotted onto a polyvinylidene difluoride (PVDF) membrane (Bio-Rad, Hercules, CA, USA), The membrane was incubated overnight at 4 °C with primary Abs, such as anti-AhR, anti-ALOX5, and anti-ALOX5AP/FLAP, followed by incubation with peroxidase-conjugated secondary Abs for 1 hr. The membranes were developed using the enhanced chemiluminescence (ECL) system.

### WST (Water Soluble Tetrazolium Salt) assay

To test cell viability, IS-trained macrophages were re-stimulated with LPS for 24 hr. Culture media was changed with serum-free RPMI medium and the WST reagent (EZ-CYTOX, DoGenBio, Seoul, Korea) followed by incubation for 1–2 hr. Measurement of the optical density value (450 nm) was performed by Infinite M200 (Tecan).

### Mouse in vivo studies

For in vivo experiments, C57BL/6 mice (7–8 weeks) were injected intraperitoneally with 200 mg/kg IS in 100 µl PBS daily for 5 days. Another 5 days after IS injection, 5 mg/kg LPS (Sigma-Aldrich) were injected intraperitoneally 75 min prior to sacrifice. Whole blood was incubated at RT for 30 min and centrifuged at $3000 \times g$ for 10 min at 4°C to collect mouse serum. The amount of TNF-α and IL-6 in serum was quantified using commercial mouse ELISA kits (Thermo Fisher Scientific). For ex vivo experiments using splenic myeloid cells, IS-trained mice were sacrificed and their spleens were aseptically collected. Single-cell splenic suspensions were prepared in PBS after passage through a 40 mm cell strainer. Splenocytes were seeded at $1 \times 10^7$ cells/well in 12-well plates. After incubation for 1 h, adherent cells were harvested for immunoblot analysis or stimulated with 10 ng/ml LPS for 24 hr. The amount of TNF-α and IL-6 in culture supernatants was quantified using commercial mouse ELISA kits (Thermo Fisher Scientific).

### Statistical analysis

A two-tailed paired or unpaired non-parametric t-test was performed to analyze data using Prism 8 (GraphPad Software, La Jolla, CA, USA) and Microsoft Excel 2013. p values of less than 0.05 were considered statistically significant.

### Study approval

Study protocols were reviewed and approved by the IRB (institutional review board) of Seoul National University Hospital and Severance Hospital. Peripheral blood of ESRD patients and HCs was drawn after obtaining written, informed consent. The methods were performed in accordance with the approved guidelines.

## Acknowledgements

The authors thank the Core Lab, Clinical Trials Center, Seoul National University Hospital for drawing blood. This work was supported in part by a grant (Grant no: 2022R1A4A1033767 and 2022R1A2C3011243 to WW Lee) from the National Research Foundation of Korea (NRF) funded by Ministry of Science and ICT (MSIT) and by a grant (Grant no: RS-2023-00238632 to HYK) of Basic Science Research Program through NRF funded by the Ministry of Education, Republic of Korea.

## Additional information

### Funding

| Funder | Grant reference number | Author |
| --- | --- | --- |
| National Research Foundation of Korea | 2022R1A4A1033767 | Won-Woo Lee |
| National Research Foundation of Korea | 2022R1A2C3011243 | Won-Woo Lee |
| National Research Foundation of Korea | RS-2023-00238632 | Hee Young Kim |

The funders had no role in study design, data collection and interpretation, or the decision to submit the work for publication.

### Author contributions

Hee Young Kim, Conceptualization, Data curation, Formal analysis, Funding acquisition, Validation, Investigation, Visualization, Methodology, Writing – original draft, Writing – review and editing; Yeon

Jun Kang, Dong Hyun Kim, Jiyeon Jang, Investigation; Su Jeong Lee, Gwanghun Kim, Formal analysis; Hee Byung Koh, Ye Eun Ko, Hajeong Lee, Tae-Hyun Yoo, Resources, Investigation; Hyun Mu Shin, Resources, Formal analysis, Investigation; Won-Woo Lee, Conceptualization, Formal analysis, Supervision, Funding acquisition, Writing – original draft, Project administration, Writing – review and editing

### Author ORCIDs
Hee Young Kim ⓘ https://orcid.org/0000-0003-3578-7344
Dong Hyun Kim ⓘ http://orcid.org/0000-0002-2765-4376
Won-Woo Lee ⓘ https://orcid.org/0000-0002-5347-9591

### Ethics
The study protocols were reviewed and approved by the institutional review board of Seoul National University Hospital (Seoul, South Korea: No. 2012-157-1184, 1403-049-564, 1306-002-491, 1109-055-378) and Severance Hospital (Seoul, South Korea: No. 4-2022-0818). Peripheral blood of ESRD patients and healthy controls (HCs) was drawn after obtaining written, informed consent. The methods were performed in accordance with the approved guidelines.

All mice were housed and maintained in a pathogen-free facility at Seoul National University (SNU) College of Medicine. All experiments were approved by the SNU Institutional Animal Care and Use Committee (Permission ID: SNU-210430-2-4).

Reviewer #1 (Public Review): https://doi.org/10.7554/eLife.87316.3.sa1
Author response https://doi.org/10.7554/eLife.87316.3.sa2

---

## Additional files

### Supplementary files
• MDAR checklist

### Data availability
Sequencing data have been deposited in GEO under accession codes GSE263019 and GSE263024. All data generated or analysed during this study are included in the manuscript and supporting files.

The following datasets were generated:

| Author(s) | Year | Dataset title | Dataset URL | Database and Identifier |
|---|---|---|---|---|
| Kim HY, Lee WW | 2024 | Uremic toxin indoxyl sulfate induces trained immunity via the AhR-dependent arachidonic acid pathway in end-stage renal disease [ChIPseq] | https://www.ncbi.nlm.nih.gov/geo/query/acc.cgi?acc=GSE263019 | NCBI Gene Expression Omnibus, GSE263019 |
| Kim HY, Lee WW | 2024 | Uremic toxin indoxyl sulfate induces trained immunity via the AhR-dependent arachidonic acid pathway in end-stage renal disease [RNAseq] | https://www.ncbi.nlm.nih.gov/geo/query/acc.cgi?acc=GSE263024 | NCBI Gene Expression Omnibus, GSE263024 |

The following previously published dataset was used:

| Author(s) | Year | Dataset title | Dataset URL | Database and Identifier |
|---|---|---|---|---|
| Kim HY, Lee WW | 2020 | Transcriptional profiling of human monocytes separated from patients with end-stage renal disease (ESRD) compared to healthy control (HC) | https://www.ncbi.nlm.nih.gov/geo/query/acc.cgi?acc=GSE155326 | NCBI Gene Expression Omnibus, GSE155326 |

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
