## [Editor Report · eLife assessment]

The authors expand the concept of a new layer to training immunity, which is currently being highlighted by several colleagues in the field. The work provides **important** hints to understand end-stage renal disease. Overall, the rational approach leads to experimental results that are **solid**.

---

## [Referee Report · Reviewer #1 (Public Review)]

In this study, Kim et al. investigated the mechanism by which uremic toxin indoxyl sulfate (IS) induces trained immunity, resulting in augmented pro-inflammatory cytokine production such as TNF and IL-6. The authors claim that IS treatment induced epigenetic and metabolic reprogramming, and the aryl hydrocarbon receptor (AhR)-mediated arachidonic acid pathway is required for establishing trained immunity in human monocytes. They also demonstrated that uremic sera from end-stage renal disease (ESRD) patients can generate trained immunity in healthy control-derived monocytes.

These are interesting results that introduce the important new concept of trained immunity and its importance in showing endogenous inflammatory stimuli-induced innate immune memory. Additional evidence proposing that IS plays a critical role in the initiation of inflammatory immune responses in patients with CKD is also interesting and a potential advance of the field.

Comments on the revised version:

In the revised manuscripts, the authors have addressed essentially almost all of the points raised by the reviewers and have revised the manuscript accordingly. The additional comments improved the manuscript and strengthened the overall impact of the paper.

---

## [Author Response]

The following is the authors’ response to the original reviews.

**Public Reviews:**

**Reviewer #1 (Public Review):**
In this study, Kim et al. investigated the mechanism by which uremic toxin indoxyl sulfate (IS) induces trained immunity, resulting in augmented pro-inflammatory cytokine production such as TNF and IL6. The authors claim that IS treatment induced epigenetic and metabolic reprogramming, and the aryl hydrocarbon receptor (AhR)-mediated arachidonic acid pathway is required for establishing trained immunity in human monocytes. They also demonstrated that uremic sera from end-stage renal disease (ESRD) patients can generate trained immunity in healthy control-derived monocytes.These are interesting results that introduce the important new concept of trained immunity and its importance in showing endogenous inflammatory stimuli-induced innate immune memory. Additional evidence proposing that IS plays a critical role in the initiation of inflammatory immune responses in patients with CKD is also interesting and a potential advance of the field. This study is in large part well done, but some components of the study are still incomplete and additional efforts are required to nail down the main conclusions.

Thank you very much for your positive feedback.

Specific comments:(1) Of greatest concern, there are concerns about the rigor of these experiments, whether the interpretation and conclusions are fully supported by the data. (1) Although many experiments have been sporadically conducted in many fields such as epigenetic, metabolic regulation, and AhR signaling, the causal relationship between each mechanism is not clear. (2) Throughout the manuscript, no distinction was made between the group treated with IS for 6 days and the group treated with the second LPS (addressed below). (3) Besides experiments using non-specific inhibitors, genetic experiments including siRNA or KO mice should be examined to strengthen and justify central suggestions.

We are grateful for the invaluable constructive feedback provided.

(1) In response to the reviewer's feedback, we conducted additional experiments employing appropriate inhibitors to investigate the causal relationship among the AhR pathway, epigenetic modifications, and metabolic rewiring in IS-induced trained immunity. Notably, metabolic rewiring, particularly the upregulation of aerobic glycolysis via the mTORC1 signaling pathway, stands as a pivotal mechanism underlying the induction of trained immunity through the modulation of epigenetic modifications (Riksen NP et al. Figure 1). Initially, we assessed the enrichment of H3K4me3 at 6-day on promoters of TNFA and IL6 loci after treatment of zileuton, an inhibitor of ALOX5, and 2-DG, a glycolysis inhibitor. Additionally, we evaluated the alteration in the activity of S6K, a downstream molecule of mTORC1, following zileuton treatment. Our findings indicate that AhR-dependent arachidonic acid (AA) signaling induces epigenetic modifications, albeit without inducing metabolic rewiring, in IS-induced trained immunity (Author response image 1). However, IS stimulation promotes mTORC1-mediated glycolysis in an AhR-independent manner. Notably, inhibition of glycolysis with 2-DG impacts epigenetic modifications. We have updated Figure 7 of the revised manuscript to incorporate these additional experimental findings, elucidating the correlation between the diverse mechanisms implicated in IS-induced innate immune memory (Fig. 7 in the revised manuscript). These data have been integrated into the revised manuscript as Figure 3D and 5I, and supplementary Figure 5I.

(2) We apologize for any confusion arising from the unclear description regarding the distinction between the group treated with IS for 6 days and the group subjected to secondary lipopolysaccharide (LPS) stimulation. It is imperative to clarify that induction of trained immunity necessitates 1 day of IS stimulation followed by 5 days of rest, rendering the 6th day sample representative of a trained state. Subsequent to this, a 24-hour LPS stimulation is applied, designating the 7th day sample as a secondary LPS-stimulated cell. This clarification is now explicitly indicated throughout the entirety of Figure 1A and Figure 3A in the revised manuscript.

(3) In accordance with your feedback, we performed siRNA knockdown of AhR and ALOX5 in primary human monocytes. AhR knockdown markedly attenuated the mRNA expression of TNF-α and IL-6, which are augmented in IS-trained macrophages. Similarly, knockdown of ALOX5 using ALOX5 siRNA abrogated the increase in TNF-α and IL-6 levels upon LPS stimulation in IS-trained macrophages (Author response image 2). Our experiments utilizing AhR siRNA corroborate the involvement of AhR in the expression of AA pathway-related molecules, such as ALOX5, ALOX5AP, and LTB4R1, in IS-induced trained immunity. These data have been incorporated into the revised manuscript as Figure 4E and 5G, and supplementary Figure 5H.

**Author response image 1. sa2fig1:** Epigenetic modification is regulated by arachidonic acid (AA) pathway and metabolic rewiring, but metabolic rewiring is not affected by the AA pathway. (**A-B**) Monocytes were pre-treated with zileuton (ZLT), an inhibitor of ALOX5, or 2DG, a glycolysis inhibitor, followed by stimulation with IS for 24 hours. After a resting period of 5 days, the enrichment of H3K4me3 on the promoters of TNFA and IL6 loci was assessed. Normalization was performed using 2% input. (**C**) Monocytes were pre-treated with zileuton (ZLT) and stimulated with IS for 24 hr. Cell lysates were immunoblotted for phosphorylated S6 Kinase, with β-actin serving as a normalization control. Band intensities in the immunoblots were quantified using densitometry. (**D**) A schematic representation of the mechanistic framework underlying IS-trained immunity. Bar graphs show the mean ± SEM. * = p < 0.05, ** = p < 0.01, and *** = p < 0.001 by two-tailed paired t-test.

**Author response image 2. sa2fig2:** Inhibition of IS-trained immunity by knockdown of AhR or ALOX5 in human monocytes. (**A-C**) Human monocytes were transfected with siRNA targeting AhR (siAhR), ALOX5 (siALOX5), or negative control (siNC) for 1 day, followed by stimulation with IS for 24 hours. After a resting period of 5 days, cells were re-stimulated with LPS for 24 hours. mRNA expression levels of AhR and ALOX5 at 1 day after transfection, and TNF-α and IL-6 at 1 day after LPS treatment, were assessed using RT-qPCR. (**D**) Human monocytes were transfected with AhR siRNA or negative control (NC) siRNA for 1 day, followed by stimulation with IS for 24 hours. After resting for 5 days, mRNA expression levels of ALOX5, ALOX5AP, and LTB4R1 were analyzed using RT-qPCR. Bar graphs show the mean ± SEM. * = p < 0.05, ** = p < 0.01, and *** = p < 0.001 by two-tailed paired t-test.

(2) The authors showed that IS-trained monocytes showed no change in TNF or IL-6, but increased the expression levels of TNF and IL-6 in response to the second LPS (Fig. 1B). This suggests that the different LPS responsiveness in IS-trained monocytes caused altered gene expression of TNF and IL6. However, the authors also showed that IS-trained monocytes without LPS stimulation showed increased levels of H3K4me3 at the TNF and IL-6 loci, as well as highly elevated ECAR and OCR, leading to no changes in TNF and IL-6. Therefore, it is unclear why or how the epigenetic and metabolic states of IS-trained monocytes induce different LPS responses. For example, increased H3K4me3 in HK2 and PFKP is important for metabolic rewiring, but why increased H3K4me3 in TNF and IL6 does not affect gene expression needs to be explained.

We acknowledge the constructive critiques provided by the reviewer. While epigenetic modifications in the promoters of TNF-α, IL-6, HK2, and PFKP (Figure 3B and Supplementary Figure 3C in the revised manuscript), and metabolic rewiring (Figure 2A-D in the revised manuscript) were observed in IS-trained macrophages at 6 days prior to LPS stimulation, these macrophages do not exhibit an increase in TNF-α and IL-6 mRNA and protein levels before LPS stimulation. This lack of response is attributed to a 5-day resting period, allowing the macrophages to revert to a non-activated state, as depicted in Author response image 3 and 4. This phenomenon aligns with the concept of typical trained immunity.

Trained immunity is characterized by the long-term functional reprogramming of innate immune cells, which is evoked by various primary insults and which leads to an altered response towards a second challenge after the return to a non-activated state. Metabolic and epigenetic reprogramming events during the primary immune response persist partially even after the initial stimulus is removed. Upon a secondary challenge, trained innate immune cells exhibit a more robust and more prompt response than the initial response (Netea MG et al. Defining trained immunity and its role in health and disease. Nat Rev Immunol. 2020 Jun;20(6):375-388).

Numerous studies have demonstrated the observation of epigenetic modifications in the promoters of TNF-α and IL-6, and metabolic rewiring prior to LPS stimulation as a secondary challenge. However, cytokine production is contingent on LPS stimulation (Arts RJ et al. Glutaminolysis and Fumarate Accumulation Integrate Immunometabolic and Epigenetic Programs in Trained Immunity. Cell Metab. 2016 Dec 13;24(6):807-819; Arts RJW et al. Immunometabolic Pathways in BCG-Induced Trained Immunity. Cell Rep. 2016 Dec 6;17(10):2562-2571; Ochando J et al. Trained immunity - basic concepts and contributions to immunopathology. Nat Rev Nephrol. 2023 Jan;19(1):23-37). The prolonged presence of higher levels of H3K4me3 on immune gene promoters, even after returning to baseline, is associated with open chromatin and results in a more rapid and stronger response, such as cytokine production, upon a secondary insult (Netea MG et al. Defining trained immunity and its role in health and disease. Nat Rev Immunol. 2020 Jun;20(6):375-388).

The results in Figure 1B may be interpreted as indicating different LPS responsiveness in IStrained monocytes caused altered gene expression of TNF and IL-6. However, it is plausible that trained immune cells respond more robustly even to low concentrations of LPS. In fact, the aim of this experiment was to determine the appropriate LPS concentration.

**Author response image 3. sa2fig3:** The changes in mRNA and protein level of TNF-α and IL-6 during induction of IS-trained immunity. Human monocytes were treated with or without IS (1 mM) for 24 hrs, succeeded by 5-day resting period to induce trained immunity. Cells were stimulated with LPS for 24 hrs. Protein and mRNA levels were assessed by ELISA and RT-qPCR, respectively. Bar graphs show the mean ± SEM. * = p < 0.05 and ** = p < 0.01, by two-tailed paired t-test.

**Author response image 4. sa2fig4:** The changes in mRNA of HK2 and PFKP induced by IS during induction of IS-trained immunity. Human monocytes were treated with or without IS (1 mM) for 24 hrs, succeeded by 5-day resting period to induce trained immunity. mRNA levels were assessed by RT-qPCR. Bar graphs show the mean ± SEM. * = p < 0.05 by two-tailed paired ttest.

(3) The authors used human monocytes cultured in human serum without growth factors such as MCSF for 5-6 days. When we consider the short lifespan of monocytes (1-3 days), the authors need to explain the validity of the experimental model.

We appreciate the reviewer’s constructive critiques. As pointed out by the reviewer, human circulating CD14+ monocytes exhibit a relatively short lifespan (1-3 days) when cultured in the absence of growth factors (Patel AA et al. The fate and lifespan of human monocyte subsets in steady state and systemic inflammation. J Exp Med. 2017 Jul 3;214(7):1913-1923). In this study, purified CD14+ monocytes were subjected to adherent culture for a duration of 7 days in RPMI1640 media supplemented with 10% human AB serum, a standard in vitro culture protocol widely employed in studies focusing on trained immunity (Domínguez-Andrés J et al. In vitro induction of trained immunity in adherent human monocytes. STAR Protoc. 2021 Feb 24;2(1):100365). In response to the reviewer's suggestions, we assessed cell viability on days 0, 1, 4, and 6, utilizing the WST assay. Despite a marginal reduction in cell viability observed at day 1, attributed to detachment from the culture plate, the cultured monocytes exhibited a notable enhancement in cell viability on days 4 and 6 when compared to days 0 or 1 (Author response image 5).

It has been demonstrated that the adhesion of human monocytes to a cell culture dish leads to their activation and induces the synthesis of substantial amounts of IL-1β mRNA as observed in monocytes adherent to extracellular matrix components such as fibronectin and collagen.

Morphologically, human adherent monocytes cultured with 10% human serum appear to undergo partial differentiation into macrophages by day 6, potentially explaining the observed lack of decrease in monocyte viability. Notably, Safi et al. have reported that adherent monocytes cultured with 10% human serum exhibit no significant difference in cell viability over a 7-day period when compared to cultures supplemented with growth factors such as M-CSF and IL-3 (Safi W et al. Differentiation of human CD14+ monocytes: an experimental investigation of the optimal culture medium and evidence of a lack of differentiation along the endothelial line. Exp Mol Med. 2016 Apr 15;48(4):e227).

**Author response image 5. sa2fig5:** Viability of human monocytes during the induction of trained immunity. Purified human monocytes were seeded on plates with RPIM1640 media supplemented with 10% human AB serum. Cell viability was assessed on days 0, 1, 4, and 6 utilizing the WST assay (Left panel). Cell morphology was examined under a light-inverted microscope at the indicated times (Right panel).

(4) The authors' ELISA results clearly showed increased levels of TNF and IL-6 proteins, but it is well established that LPS-induced gene expression of TNF and IL-6 in monocytes peaked within 1-4 hours and returned to baseline by 24 hours. Therefore, authors need to investigate gene expression at appropriate time points.

We appreciate the valuable constructive feedback provided by the reviewer. As indicated by the reviewer, the LPS-induced gene expression of TNF-α and IL-6 in IS-trained monocytes exhibited a peak within the initial 1 to 4 hours, followed by a decrease by the 24-hour time point, as illustrated in Author response image 6. Nevertheless, the mRNA expression levels of TNFα and IL-6 were still elevated at the 24-hour mark. Furthermore, the protein levels of both TNFα and IL-6 apparently increased 24 hours after LPS stimulation. Due to technical constraints, sample collection had to be conducted at a single time point, and the 24-hour post-stimulation interval was deemed optimal for this purpose.

**Author response image 6. sa2fig6:** Kinetics of protein and mRNA expression of TNF-α and IL-6 after treatment of LPS as secondary insult in IS-trained monocytes. IS-trained cells were re-stimulated by LPS (10 ng/ml) for the indicated time. The supernatant and lysates were collected for ELISA assay and RT-qPCR analysis, respectively. Bar graphs show the mean ± SEM. * = p <0.05 and ** = p < 0.01, by two-tailed paired t-test.

(5) It is a highly interesting finding that IS induces trained immunity via the AhR pathway. The authors also showed that the pretreatment of FICZ, an AhR agonist, was good enough to induce trained immunity in terms of the expression of TNF and IL-6. However, from this point of view, the authors need to discuss why trained immunity was not affected by kynurenic acid (KA), which is a well-known AhR ligand accumulated in CKD and has been reported to be involved in innate immune memory mechanisms (Fig. S1A).

We appreciate the constructive criticism provided by the reviewer, and we comprehend the raised points. In our initial experiments, we hypothesized that kynurenic acid (KA), an aryl hydrocarbon receptor (AhR) ligand, might instigate trained immunity in monocytes, despite KA not being our primary target uremic toxin. However, our findings, as depicted in Fig. S1A, demonstrated that KA did not induce trained immunity. Notably, KA-treated monocytes exhibited induction of CYP1B1, an AhR-responsive gene, and elevated levels of TNF-α and IL-6 mRNA at 24 hours post-treatment, comparable to FICZ-treated monocytes. This observation underscores KA's role as an AhR ligand in human monocytes, as emphasized by the reviewer.

Of particular interest, proteins associated with the arachidonic acid pathway, such as ALOX5 and ALOX5AP - integral to the mechanisms underlying IS-induced trained immunity - did not exhibit an increase at day 6 following KA treatment, in contrast to the significant elevation observed with IS and FICZ treatments (Author response image 7). The rationale behind this disparity remains unknown, necessitating further investigation to elucidate the underlying factors. These data have been incorporated into the revised manuscript as Supplementary Figure 5C.

**Author response image 7. sa2fig7:** Divergent impact of AhR agonists, especially IS, FICZ, and KA on the AhR-ALOX5 pathway. Purified ytes underwent treatment with IS (1 mM), FICZ (100 nM), or KA (0.5 mM) for 1 day, followed by 5-day resting period to trained immunity. Activation of AhR through ligand binding was assessed by examining the induction of CYP1B1, an AhR ene, and cytokines one day post-treatment. The expression of genes related to the arachidonic acid pathway, such as ALOX5, 5AP, and LTB4R1, was analyzed via RT-qPCR six days after inducing trained immunity. Bar graphs show the mean ± SEM. * .05, ** = p < 0.01, and *** = p < 0.001 by two-tailed paired t-test.

Indeed, it has been demonstrated that FICZ and TCDD, two high-affinity AhR ligands, exert opposite effects on T-cell differentiation, with TCDD inducing regulatory T cells and FICZ inducing Th17 cells. This dichotomy has been attributed to ligand-intrinsic differences in AhR activation (Ho PP et al. The aryl hydrocarbon receptor: a regulator of Th17 and Treg cell development in disease. Cell Res. 2008 Jun;18(6):605-8; Ehrlich AK et al. TCDD, FICZ, and Other High Affinity AhR Ligands Dose-Dependently Determine the Fate of CD4+ T Cell Differentiation. Toxicol Sci. 2018 Feb 1;161(2):310-320). These outcomes imply the involvement of an intricate interplay involving metabolic rewiring, epigenetic reprogramming, and the AhR-ALOX5 pathway in IS-induced trained immunity within monocytes.

(6) The authors need to clarify the role of IL-10 in IS-trained monocytes. IL-10, an anti-inflammatory cytokine that can be modulated by AhR, whose expression (Fig. 1E, Fig. 4D) may explain the inflammatory cytokine expression of IS-trained monocytes.

We appreciate the reviewer’s valuable comment, recognizing its significant importance. IL-10, characterized by potent anti-inflammatory attributes, assumes a pivotal role in constraining the host immune response against pathogens. This function serves to mitigate potential harm to the host and uphold normal tissue homeostasis. In the context of atherosclerosis (Mallat Z et al. Protective role of interleukin-10 in atherosclerosis. Circ Res. 1999 Oct 15;85(8):e17-24.) and kidney disease (Wei W et al. The role of IL-10 in kidney disease. Int Immunopharmacol. 2022 Jul;108:108917), IL-10 exerts potent deactivating effects on macrophages and T cells, influencing various cellular processes that could impact the development and stability of atherosclerotic plaques. Additionally, it is noteworthy that IL-10-deficient macrophages exhibit an augmentation in the proinflammatory cytokine TNF-α (Smallie T et al. IL-10 inhibits transcription elongation of the human TNF gene in primary macrophages. J Exp Med. 2010 Sep 27;207(10):2081-8; Couper KN et al. IL-10: the master regulator of immunity to infection. J Immunol. 2008 May 1;180(9):5771-7). As emphasized by the reviewer, the reduced gene expression of IL-10 by IS-trained monocytes may contribute to the heightened expression of proinflammatory cytokines. We have thoroughly addressed and discussed this specific point in response to the reviewer's comment (Line 394-399 of page 18 in the revised manuscript).

(7) The authors need to show H3K4me3 levels in TNF and IL6 genes in all conditions in one figure. (Fig. 2B). Comparing Fig. 2B and Fig. S2B, H3K4me3 does not appear to be increased at all by LPS in the IL6 region.

We are grateful for the constructive criticism provided by the reviewer. In response to the reviewer's comment, we endeavored to conduct an experiment demonstrating H3K4me3 enrichment on the promoters of TNF-α and IL-6 across all experimental conditions. However, due to limitations in the availability of purified human monocytes, we conducted an additional three independent experiments for ChIP-qPCR across all conditions. Despite encountering a notable variability among individuals, even within the healthy donor cohort, our results demonstrated an increase in H3K4me3 enrichment on the TNF-α and IL-6 promoters in IS-trained groups, irrespective of subsequent LPS treatment (Author response image 8).

**Author response image 8. sa2fig8:** Analysis of H3K4me3 enrichment on the promoters of TNFA and IL6 Loci in IS-trained macrophages. ChIP-qPCR was employed to assess the enrichment of H3K4me3 on the promoters of TNFA and IL6 loci before (day 6) and after LPS stimulation (day 7) in IS-trained macrophages. The normalization control utilized 2% input. Bar graphs show the mean ± SEM. The data presented are derived from three independent experiments utilizing samples from different donors.

(8) The authors need to address the changes of H3K4me3 in the presence of MTA.

We appreciate the constructive criticism provided by the reviewer. In response to the reviewer's feedback, we conducted an analysis of the changes in H3K4me3 in the presence of MTA, a general methyltransferase inhibitor, using identical conditions as depicted in Figure 2C of the original manuscript. Our findings revealed that MTA exerted inhibitory effects on the levels of H3K4me3, as isolated through the acid histone extraction method, which were otherwise increased by IS-training, as illustrated in Author response image 9.

**Author response image 9. sa2fig9:** The reduction of H3K4me3 by MTA treatment in IS-trained macrophages. IS-trained cells were restimulated by LPS (10 ng/ml) as a secondary challenge for 24 hrs, followed by isolation of histone and WB analysis for H3K4me3, Histone 3 (H3), and β-actin. The blot data from two independent experiments with different donors were shown.

(9) Interpretation of ChIP-seq results is not entirely convincing due to doubts about the quality of sequencing results. First, authors need to provide information on the quality of ChIP-seq data in reliable criteria such as Encode Pipeline. It should also provide representative tracks of H3K4me3 in the TNF and IL-6 genes (Fig. 2F). And in Fig. 2F, the author showed the H3K4me3 track of replicates, but the results between replicates were very different, so there are concerns about reproducibility. Finally, the authors need to show the correlation between ChIP-seq (Fig. 2) and RNA-seq (Fig. 5).

We appreciate the constructive criticism provided by the reviewer.

As indicated by the reviewer, for evaluation of sample read quality, analysis was performed using the histone ChIP-seq standard from the ENCODE project, focusing on metrics such as read depth, PCR bottleneck coefficient (PBC)1, PBC2, and non-redundant fraction (NRF). Five of the total samples were displayed moderate bottleneck levels (0.5 ≤ PBC1 < 0.8, 1 ≤ PBC2 < 3) with acceptable (0.5 ≤ NRF < 0.8) complexity. One sample showed mild bottlenecks (0.8 ≤ PBC1 < 0.9, 3 ≤ PBC2 < 10) with compliance (0.8 ≤ NRF < 0.9) complexity. This quality metrics indicated ChIP-seq data quality meets at least the standards required for downstream analysis according to ENCODE project criteria (Author response image 10A).

To examine the differences in H3K4me3 enrichment patterns between two groups, we normalized the read counts around the TSS ±2 kb of human genes to CPM. Sequentially, we compared the average values of IS-treated macrophage compare to control and displayed in waterfall plots. In addition, we marked genes of interest in red including the phenotypes of IStrained macrophages (TNF and IL6), the activation of the innate immune responses (XRCC5, IFI16, PQBP1), and the regulation of ornithine decarboxylase (OAZ3, PSMA3, PSMA1) (Author response image 10B and C). Also, H3K4me3 peak tracks of TNF and IL6 loci and H3K4me3 enrichment pattern were added in supplementary Figure 3D and 3F in the revised manuscript.

Next, to evaluate the consistency among replicates within a group, we analyzed enrichment values, expressed as Counts per Million (CPM) using edgeR R-package, by applying Spearman's correlation coefficients. we analyzed two sets included total 7,136 H3K4me3 peak sets, as described in Figure 3E in the revised manuscript and 2 kbp around transcription start sites (TSS) from hg19 human genomes. The resulting Spearman's correlation coefficients and associated P-values demonstrated a concordance between replicates, confirming reproducibility and consistent performance (Author response image 10D).

Finally, the correlation between gene expression and H3K4me3 enrichment around transcription start sites (TSS) has been reported in previous research (Reshetnikov VV et al. Data of correlation analysis between the density of H3K4me3 in promoters of genes and gene expression: Data from RNA-seq and ChIP-seq analyses of the murine prefrontal cortex. Data Brief. 2020 Oct 2;33:106365). To verify this association in our study, we applied Spearman's correlation for comparative analysis and conducted linear regression to determine if a consistent global trend in RNA expression existed. In our analysis, count values from regions extending 2 kbp around the TSSs in H3K4me3 ChIP-seq data were converted to Counts per Million (CPM) using edgeR R-package. These were then contrasted with the Transcripts Per Million (TPM) values of genes. Our results revealed a significant positive correlation, reinforcing the consistent relationship between H3K4me3 enrichment and gene expression (Author response image 10E and Supplementary Fig. 6D in revised manuscripts).

**Author response image 10. sa2fig10:** The information on quality of ChIP-seq data and correlation between ChIP-seq and RNA-seq. (**A**) Information on quality of ChIP-seq data. (**B**) H3K4me3 peak of promoter region on TNFA and IL6. (**C**) The differences in H3K4me3 enrichment patterns between control group and IS-training group. (**D**) The consistency among replicates within a group. (**E**) Correlation between ChIP-seq and RNA-seq in IS-induced trained immunity.

(10) AhR changes in the cell nucleus should be provided (Fig. 4A).

We appreciate the constructive feedback from the reviewer. In response to the reviewer's suggestions, we investigated the nuclear translocation of AhR on 6 days after the induction of ISmediated trained immunity, as illustrated in Author response image 11. For this purpose, the lysate from IS-trained monocytes was fractionated into the nucleus and cytosol, and AhR protein was subsequently immunoblotted. The results depicted in Figure X demonstrate that IS-trained monocytes exhibited a higher level of AhR protein in the nucleus compared to non-trained monocytes. Notably, the nuclear translocation of AhR was significantly attenuated in IS-trained monocytes treated with GNF351. These findings imply that the activation of AhR, facilitated by the binding of IS, persisted partially up to 6 days, indicating that IS-mediated degradation of AhR was not fully recovered even on day 6 after the induction of IS training. Consequently, we have replaced Figure 4A in the revised manuscript.

**Author response image 11. sa2fig11:** The activation of AhR, facilitated by IS binding, is persisted partially up to 6 days during induction of trained immunity. The lysate of IS-trained cells treated with or without GNF351, were separated into nuclear and cytosol fraction, followed by WB analysis for AhR protein (Left panel). Band intensity in immunoblots was quantified by densitometry (Right panel). β-actin was used as a normalization control. Bar graphs show the mean ± SEM. * = p < 0.05, by two-tailed paired t-test.

(11) Do other protein-bound uremic toxins (PBUTs), such as PCS, HA, IAA, and KA, change the mRNA expression of ALOX5, ALOX5AP, and LTB4R1? In the absence of genetic studies, it is difficult to be certain of the ALOX5-related mechanism claimed by the authors.

We are grateful for the constructive criticism provided by the reviewer. In response to the reviewer's comment, we investigated whether uremic toxins, specifically PBUTs such as PCS, HA, IAA, and KA, induce changes in the mRNA expression of ALOX5, ALOX5AP, and LTB4R1 in trained monocytes. Intriguingly, the examination revealed no discernible induction in the mRNA expression of these genes by PBUTs, with the exception of IS, as depicted in Author response image 12 of the letter. These findings once again underscore the implication of the AhR-ALOX5 pathway in the induction of trained immunity in monocytes by IS.

**Author response image 12. sa2fig12:** No obvious impact of PBUTs except IS on the expression of arachidonic acid pathway-related genes on 6 days after treatment with PBUTs. Purified monocytes were treated with several PBUTs including IS, PCS, HA, IAA, and KA for 24 hrs., following by 5-day resting period to induce trained immunity. The mRNA expression of ALOX5, ALOX5AP, and LTB4R1 were quantified using RT-qPCR. Bar graphs show the mean ± SEM. * = p < 0.05, by two-tailed paired t-test.

(12) Fig.6 is based on the correlated expression of inflammatory genes or AA pathway genes. It does not clarify any mechanisms the authors claimed in the previous figures.

We express our sincere appreciation for the constructive criticism provided by the reviewer, and we have taken careful note of the points raised. In response to the reviewer's feedback, we adopted two distinct approaches utilizing samples obtained from ESRD patients and IS-trained mice. Initially, we investigated the correlation between ALOX5 protein expression in monocytes and IS concentration in the plasma of ESRD patients presented in Figure 6E of the original manuscript. Despite the limited number of samples, our analysis revealed a nonsignificant correlation between IS concentration and ALOX5 expression; however, it demonstrated a positive trend (Author response image 13A). Subsequently, we examined the potential inhibitory effects of zileuton, an ALOX5 inhibitor, on the production of TNF-α and IL-6 in LPSstimulated splenic myeloid cells derived from IS-trained mice. Our findings indicate that zileuton significantly inhibits the production of TNF-α and IL-6 induced by LPS in splenic myeloid cells from IS-trained mice (Author response image 13B). These data were added in Figure 6N of the revised manuscript (Line 350-354 of page 16 in the revised manuscript).

**Author response image 13. sa2fig13:** Assessment of the correlation between ALOX5 and the concentration of IS in ESRD patients, and investigation of ALOX5 effects in mouse splenic myeloid cells in IS-trained mice. (**A**) Examination of the correlation between ALOX5 protein expression in monocytes and IS concentration in the plasma of ESRD patients. (**B**) C57BL/6 mice were administered daily injections of 200 mg/kg IS for 5 days, followed by a resting period of another 5 days. Subsequently, IS-trained mice were sacrificed, and spleens were mechanically dissociated. Isolated splenic myeloid cells were subjected to ex vivo treatment with LPS (10 ng/ml), along with zileuton (100 µM). The levels of TNF-α and IL-6 in the supernatants were quantified using ELISA. The graphs show the mean ± SEM. * = p < 0.05, by two-tailed paired t-test between zileuton treatment group and no-treatment group.

**Recommendations for the authors:**

**Reviewer #1 (Recommendations For The Authors):**
Minor corrections to the figures(1) No indicators for the control group in Fig. 1B.

We thank you for the reviewer’s comment. According to the reviewer’s comment, the control group was indicated with (-).

(2) The same paper is listed twice in the references section. (No. 19 and 28)

We thank you for the reviewer’s comment. We deleted the reference No. 28.

**Reviewer #2 (Public Review):**
Manuscript entitled "Uremic toxin indoxyl sulfate (IS) induces trained immunity via the AhR-dependent arachidonic acid pathway in ESRD" presented some interesting findings. The manuscript strengths included use of H3K4me3-CHIP-Seq, AhR antagonist, IS treated cell RNA-Seq, ALOX5 inhibitor, MTA inhibitor to determine the roles of IS-AhR in trained immunity related to ESRD inflammation and trained immunity.

Thank you very much for your positive feedback.

**Reviewer #2 (Recommendations For The Authors)**:However, the manuscript needs to be improved by fixing the following concerns.There are concerns:(1) The experiments in Figs. 1G, 1H and 1I need to have AhR siRNA, and siRNA control to demonstrate that the results in uremic toxins-containing serum-treated experiments were related to IS;

We extend our gratitude to the reviewer for their invaluable comment, acknowledging its significant relevance to our study. In accordance with the reviewer's suggestion, we endeavored to conduct additional experiments utilizing AhR siRNA to elucidate the direct impact of IS present in the serum of end-stage renal disease (ESRD) patients on the induction of IS-mediated trained immunity.

Regrettably, owing to limitations in the availability of monocytes post-siRNA transfection, we were unable to establish a direct relationship between the observed outcomes in experiments utilizing uremic toxins-containing serum and IS in AhR siRNA knockdown monocytes. However, treatment with GNF351, an AhR antagonist, resulted in the inhibition of TNF-α production in trained monocytes exposed to uremic toxins-containing serum (Author response image 14).

In our previous studies, we have already reported that uremic serum-induced TNF-α production in human monocytes is dependent on the AhR pathway, using GNF351 (Kim HY et al. Indoxyl sulfate (IS)-mediated immune dysfunction provokes endothelial damage in patients with end-stage renal disease (ESRD). Sci Rep. 2017 Jun 8;7(1):3057). Additionally, we have provided evidence demonstrating an augmentation in the activity of the AhR pathway within monocytes derived from ESRD patients, indicative of a significant reduction in AhR protein levels (Kim HY et al. Indoxyl sulfate-induced TNF-α is regulated by crosstalk between the aryl hydrocarbon receptor, NF-κB, and SOCS2 in human macrophages. FASEB J. 2019 Oct;33(10):10844-10858). It is noteworthy that other major protein-bound uremic toxins (PBUTs), such as PCS, HA, IAA, and KA, failed to induce trained immunity in human monocytes (Supplementary Figure 1A in the revised manuscript). Nevertheless, knockdown of AhR via siRNA effectively impeded the induction of IS-mediated trained immunity in human monocytes (Figure 4E in the revised manuscript).

Taken collectively, our findings suggest a critical role for IS present in the serum of ESRD patients in the induction of trained immunity in human monocytes.

**Author response image 14. sa2fig14:** Inhibition of uremic serum (US)-induced trained immunity by AhR antagonist, GNF351. Monocytes were pre-treated with or without GNF351 (AhR antagonist; 10 µM) for 1 hour, followed by treatment with pooled normal serum (NS) or uremic serum (US) at a concentration of 30% (v/v) for 24 hours. After a resting period of 5 days, cells were stimulated with LPS for 24 hours. The production of TNF-α and IL-6 in the supernatants was quantified using ELISA. The data presented are derived from three independent experiments utilizing samples from different donors.

(2) Fig. 3 needs to be moved as Fig. 2

We express appreciation for the constructive suggestion provided by the reviewer. In response to the reviewer's comment, the sequence of Figure 3 and Figure 2 was adjusted in the revised manuscript.

(3, 4) The connection between bioenergetic metabolism pathways and H3K4me3 was missing; The connection between bioenergetic metabolism pathways and ALOX5 was missing;

We appreciate the reviewer’s constructive criticism and fully understood the reviewer's points. In response to the reviewer's feedback, we conducted additional experiments employing appropriate inhibitors to elucidate the interrelation between bioenergetic metabolism and H3K4me3 and between bioenergetic metabolism and ALOX5. Initially, we assessed the enrichment of H3K4me3 at 6-day on promoters of TNFA and IL6 loci after treatment of 2-DG, a glycolysis inhibitor. Additionally, we evaluated the alteration in the activity of S6K, a downstream molecule of mTORC1, following treatment with zileuton, an inhibitor of ALOX5. Our findings indicate that AhR-dependent arachidonic acid (AA) signaling induces epigenetic modifications, albeit without inducing metabolic rewiring, in IS-induced trained immunity (Author response image 15). However, IS stimulation promotes mTORC1-mediated glycolysis in an AhR-independent manner. Notably, inhibition of glycolysis with 2-DG impacts epigenetic modifications. We have updated Figure 7 of the revised manuscript to incorporate these additional experimental findings, elucidating the correlation between the diverse mechanisms implicated in IS-induced innate immune memory (Fig. 7 in the revised manuscript).

**Author response image 15. sa2fig15:** Epigenetic modification is regulated by arachidonic acid (AA) pathway and metabolic rewiring, but metabolic rewiring is not affected by the AA pathway. (**A, B**) Monocytes were pre-treated with zileuton (ZLT), an inhibitor of ALOX5, or 2DG, a glycolysis inhibitor, followed by stimulation with IS for 24 hours. After a resting period of 5 days, the enrichment of H3K4me3 on the promoters of TNFA and IL6 loci was assessed. Normalization was performed using 2% input. (**C**) Monocytes were pre-treated with ziluton (ZLT) and stimulated with IS for 24 hr. Cell lysates were immunoblotted for phosphorylated S6 Kinase, with β-actin serving as a normalization control. Band intensities in the immunoblots were quantified using densitometry. (**D**) A schematic representation of the mechanistic framework underlying IS-trained immunity. Bar graphs show the mean ± SEM. * = p < 0.05, ** = p < 0.01, and *** = p < 0.001 by two-tailed paired t-test.

(5) It was unclear whether histone acetylations such as H3K27acetylation and H3K14 acetylation are involved in IS-induced epigenetic reprogramming or IS-induced trained immunity is highly histone methylation-specific.

We appreciate the constructive comment provided by the reviewer. As highlighted by the reviewer, alterations in epigenetic histone markers, specifically H3K4me3 or H3K27ac, have been recognized as the underlying molecular mechanism in trained immunity. Due to limitations in the availability of trained cells, this study primarily focused on histone methylation. In response to the reviewer's inquiry, we briefly investigated the impact of histone acetylation using C646, a histone acetyltransferase inhibitor, on IS-induced trained immunity (Author response image 16). Our experiments revealed that C646 treatment effectively hinders the production of TNF-α and IL-6 by IS-trained monocytes in response to LPS stimulation, comparable to the effects observed with MTA (5’methylthioadenosine), a non-selective methyltransferase inhibitor. This suggests that histone acetylation also contributes to the epigenetic modifications associated with IS-induced trained immunity. We sincerely appreciate the valuable input from the reviewer.

**Author response image 16. sa2fig16:** The role of histone acetylation in epigenetic modifications in IS-induced trained immunity. Monocytes were pretreated with MTA (methylthioadenosine, methyltransferase inhibitor) or C646 (histone acetyltransferase p300 inhibitor), followed treatment with IS 1 mM for 24 hrs. After resting for 5 days, trained cells were re-stimulated by LPS 10 ng/ml as secondary insult. TNF-α and IL-6 in supernatants were quantified by ELISA. Bar graphs show the mean ± SEM. * = p < 0.05 and ** = p < 0.01 by two-tailed paired t-test.

**Reviewer #3 (Public Review):**
The manuscript entitled, "Uremic toxin indoxyl sulfate induces trained immunity via the AhRdependent arachidonic acid pathway in ESRD" demonstrates that indoxyl sulfate (IS) induces trained immunity in monocytes via epigenetic and metabolic reprogramming, resulting in augmented cytokine production. The authors conducted well-designed experiments to show that the aryl hydrocarbon receptor (AhR) contributes to IS-trained immunity by enhancing the expression of arachidonic acid (AA) metabolism-related genes such as arachidonate 5-lipoxygenase (ALOX5) and ALOX5 activating protein (ALOX5AP). Overall, this is a very interesting study that highlights that IS mediated trained immunity may have deleterious outcomes in augmented immune responses to the secondary insult in ESRD. Key findings would help to understand accelerated inflammation in CKD or RSRD.

We greatly appreciate your positive feedback.

**Reviewer #3 (Recommendations for The Authors):**
This reviewer, however, has the following concerns.Major comments:(1) Figure 1B: IS is known to induce the expression of TNF-a and IL-6. This reviewer wonders why these molecules were not detected in the IS (+) LPS (-) condition.

We appreciate the constructive comment provided by the reviewer. In our prior investigation, it was observed that the expression of TNF-α and IL-6 was induced 24 hours after IS treatment in human monocytes and macrophages (Couper KN et al. IL-10: the master regulator of immunity to infection. J Immunol. 2008 May 1;180(9):5771-7). In adherence to the trained immunity protocol, the medium was replaced at the 24 hours post-IS treatment to eliminate IS, with a subsequent change after a 5-day resting period. Probably, TNF-α and IL-6 are accumulated and detected in the IS (+) LPS (-) culture supernatant if the media was not changed at these specific time points. Our primary objective, however, was to ascertain the role of IS in the induction of trained immunity, prompting an investigation into whether IS contributes to an increase in the production of TNF-α and IL-6 in response to LPS stimulation as a secondary insult.

(2) 1' stimulus is IS followed by 2' stimulus LPS/Pam3. It would be interesting to know what the immune profile is when other uremic toxin is used for secondary insult, this would be more relevant in clinical context of ESRD.

The reviewer's insightful comment is greatly appreciated. To address their feedback, IStrained macrophages were subjected to additional stimulation using protein-bound uremic toxins (PBUTs) as a secondary challenge. As illustrated in Letter figure 17, the examined uremic toxins, namely p-cresyl sulfate (PCS), Hippuric acid (HA), Indole 3-acetic acid (IAA), and kynurenic acid (KA), failed to elicit the production of proinflammatory cytokines, specifically TNF-α and IL-6, by IS-trained monocytes.

**Author response image 17. sa2fig17:** No obvious effect of protein-bound uremic toxin (PBUTs) as secondary insults on the production of proinflammatory cytokines in IS-trained monocytes. IS-trained monocytes were re-stimulated with several PBUTs, such as IS (1 mM), PCS (1 mM), HA (2 mM), IAA. (0.5 mM), and KA (0.5 mM) as a secondary challenge for 24 hrs. TNF-α and IL-6 in supernatants were quantified by ELISA. The data from two independent experiments with different donors were shown. ND indicates ‘not detected’.

(3) The authors need to explain a rationale why RNA and protein data used different markers.

We appreciate the constructive input provided by the reviewer. Given that TNF-α and IL6 represent prototypical cytokines synthesized by trained monocytes in humans, we conducted a comprehensive analysis of their mRNA and protein levels. In human macrophages, the release of active IL-1β necessitates a second priming event, such as the presence of ATP. Consequently, we posited that assessing the mRNA levels of IL-1β would suffice to demonstrate the induction of trained immunity in our experimental protocol. Nevertheless, in response to the reviewer's comment, we proceeded to assess the protein levels of IL-1β, IL-10, and MCP-1 as illustrated in Author response image 189. These data have been incorporated into the revised manuscript as supplementary Figure 1E.

**Author response image 18. sa2fig18:** Modulation of cytokine levels in IS-trained macrophages in response to secondary stimulation with LPS. Human monocytes were stimulated with the IS for 24 hr, followed by resting period for 5 days. On day 6, the cells were re-stimulated with LPS for 24 hr. The levels of each cytokine in the supernatants were quantified using ELISA. Bar graphs show the mean ± SEM. ** = p < 0.01 and *** = p < 0.001 by two-tailed paired t-test.

(4) Epigenetic modification primarily involves histone modification and DNA methylation. The authors presented convincing data on histone modification (Figure 2), but did not provide any insights in the promoter DNA methylation status.

We express our gratitude to the reviewer for providing valuable comments, which highlight a crucial aspect of our study. Despite the well-established primary role of DNA methylation in epigenetic modifications, recent suggestions propose that histone modifications, particularly H3K4me3 or H3K27ac, play a predominant role in the induction of trained immunity. In this context, our primary inquiry was focused on determining whether IS, as an endogenous insult, induces trained immunity in monocytes, and if so, whether IS-trained immunity is mediated through metabolic and epigenetic modifications - recognized as the major mechanisms underlying the generation of trained immunity. It is imperative to note that our study's primary objective did not encompass the identification of various epigenetic changes. In response to the reviewer's inquiry, we conducted a brief examination of the impact of DNA methylation using ZdCyd (5-aza-2’-deoxycytidine), a DNA methylation inhibitor, on IS-induced trained immunity. Our experimental findings indicate that ZdCyd treatment exerts no discernible effect on the production of TNF-α and IL-6 by IS-trained monocytes upon stimulation with LPS, as illustrated in Author response image 19. However, a recent study has shed light on the role of DNA methylation in BCG vaccine-induced trained immunity in human monocytes (Bannister S et al. Neonatal BCG vaccination is associated with a long-term DNA methylation signature in circulating monocytes. Sci Adv. 2022 Aug 5;8(31):eabn4002). Consequently, further investigations utilizing DNA methylation sequencing are warranted to elucidate whether DNA methylation is implicated in the induction of IS-trained immunity.

**Author response image 19.**

**Author response image 19. sa2fig19:** The effect of DNA methylation on IS-induced trained immunity. Monocytes were pretreated with ZdCyd (5-aza-2’-deoxycytidine, DNA methylation inhibitor), followed by treatment with IS 1 mM for 24 hrs. After resting for 5 days, cells were re-stimulated by LPS 10 ng/ml as secondary insult. TNF-α and IL-6 in supernatants were quantified byELISA. Bar graphs show the mean ± SEM. * = p < 0.05 and ** = p < 0.01 by two-tailed paired t-test.

(5) Metabolic rewiring in trained immunity cells undergo metabolic changes which involved intertwined pathways of glucose and cholesterol metabolism. The authors presented nice data on glucose pathway (Figure 3) but failed to show any changes related to cholesterol metabolism.

We express our gratitude to the reviewer for providing valuable comments, which underscore a noteworthy observation. In the current investigation, our primary emphasis has been on glycolytic reprogramming, recognized as a principal mechanism for inducing trained immunity in monocytes. This focus stems from preliminary experiments wherein Fluvastatin, a cholesterol synthesis inhibitor, demonstrated no discernible impact on TNF-α production by IS-trained monocytes, as illustrated in Author response image 20. Intriguingly, Fluvastatin treatment exhibited a partial inhibitory effect on the production of IL-6 by IS-trained monocytes. Subsequent investigations are imperative to elucidate the role of cholesterol metabolism in the induction of IS-trained immunity.

**Author response image 20. sa2fig20:** The effect of cholesterol metabolism on IS-induced trained immunity. Monocytes were pretreated with Fluvastatin (cholesterol synthesis inhibitor, HMG-CoA reductase inhibitor), followed treatment with IS 1 mM for 24 hrs. After resting for 5 days, cells were re-stimulated by LPS 10 ng/ml as secondary insult. TNF-α and IL-6 in supernatants were quantified by ELISA. Bar graphs show the mean ± SEM. * = p < 0.05 and ** = p < 0.01 by two-tailed paired t-test.

(6) Trained immunity involves neutrophils in addition to monocyte/macrophages. It is evident from the RNAseq data that neutrophil degranulation (Figure 5B) is the top enriched pathway. This reviewer wonders why the authors did not perform any assays on neutrophils.

We appreciate the reviewer for valuable comment. IS represents a major uremic toxin that accumulates in the serum of patients with chronic kidney disease (CKD), correlating with CKD progression and the onset of CKD-related complications, including cardiovascular diseases (CVD). Our prior investigations have demonstrated that IS promotes the production of TNF-α and IL-1β by human monocytes and macrophages. Additionally, macrophages pre-treated with IS exhibit a significant augmentation in TNF-α production when exposed to a low dose of lipopolysaccharide (LPS). Considering the pivotal role of proinflammatory macrophages and TNF-α, a principal cardiotoxic cytokine, in CVD pathogenesis, our focus in this study has primarily focused on elucidating the trained immunity of monocytes/macrophages. Consequently, all experiments were meticulously conducted using highly purified monocytes and monocytederived macrophages derived from both healthy controls and end-stage renal disease (ESRD) patients. The reviewer's observation regarding the potential involvement of neutrophils in trained immunity has been duly noted. Subsequent investigations will be imperative to explore the conceivable role of IS-trained neutrophils in the pathogenesis of CVD. Once again, we appreciate the reviewer for their valuable comment.

(7) Figure 5C (GSEA plots): This reviewer is not sure if one can present the plots assigned with groups (eg. IS(T) vs Control). More details are required in the Methods related to this.

We apologize for any ambiguity resulting from the previously unclear description of methods concerning Gene Set Enrichment Analysis (GSEA) plots. To provide clarification, additional details pertaining to this aspect have been explained upon in the revised manuscript's Methods section.

(8) In vivo data (Figure 6 I-M): Instead of serum profile and whole set of spleen myeloid cells, it would be interesting to see changes of markers on peritoneal macrophages or bone marrow-derived macrophages since the in vitro findings are on monocyte-derived macrophages.

We appreciate comment and the insightful suggestion provided by the reviewer. In response to the reviewer's feedback, we conducted additional in vivo experiments to examine the production of TNF-α and IL-6 in bone marrow-derived macrophages (BMDMs) derived from IStrained mice. Upon LPS stimulation, we observed an increase in the production of TNF-α and IL-6 in spleen myeloid cells from IS-trained mice. However, no such increase in these cytokines was noted in BMDMs derived from the same mice (Author response image 22, A and B). In fact, we already observed that that the expression of ALOX5 was not elevated in BM cells derived from IS-trained mice presented in Figure 6L and M of the original manuscript (Author response image 22C).

Recent studies have indicated that trained immunity can be induced in circulating immune cells, such as monocytes or resident macrophages (peripheral trained immunity), as well as in hematopoietic stem and progenitor cells (HSPCs) within the bone marrow (central trained immunity) (Kaufmann E et al. BCG Educates Hematopoietic Stem Cells to Generate Protective Innate Immunity against Tuberculosis. Cell. 2018 Jan 11;172(1-2):176-190.e19; Riksen NP et al. Trained immunity in atherosclerotic cardiovascular disease. Nat Rev Cardiol. 2023 Dec;20(12):799-811). It is plausible that central trained immunity in BM progenitor cells may not be elicited in our mouse model, which is relatively acute in nature. Further investigations are warranted to explore the role of IS in inducing central trained immunity, utilizing appropriate chronic disease models.

We have included this additional data as supplementary figures in the revised manuscript (Suppl. Fig. 7, D and E, and line 355-362 of page 16 in the revised manuscript).

**Author response image 21. sa2fig21:** Absence of trained immunity in bone marrow derived macrophages (BMDMs) derived from IStrained mice. (**A, B**) IS was intraperitoneally injected daily for 5 days, followed by training for another 5 days. Isolated BM progenitor cells and spleen myeloid cells were differentiated or treated with LPS for 24 hr. The supernatants were collected for ELISA. (**C**) The level of ALOX5 protein in BM cells isolated from IS-trained or control mice was analyzed by western blot. The graph illustrates the band intensity quantified by densitometry. Bar graphs show the mean ± SEM. * = p < 0.05 and ** = p < 0.01, by unpaired t-test.

(9) Figure 7: There are no data on signaling pathway(s) that links IS and epigenetic changes, the authors therefore may want to add "?" to the proposed mechanism.

We extend our sincere appreciation to the reviewer for providing valuable feedback. In light of the constructive comments provided by three reviewers, we have undertaken a series of additional experiments. These efforts have enabled us to propose a more elucidating schematic representation of the proposed mechanism, free of any ambiguous elements (Figure 7 in the revised manuscript). We are grateful for your insightful input.

(10) Demographic data (Table S2): ESRD patients have co-morbidities including diabetes (33% of subjects), CAD (28%). How did the authors factor out the co-morbidities in the overall context of their findings?

We express gratitude to the reviewer for providing valuable comments, particularly on a noteworthy and significant aspect. The investigation employed an End-Stage Renal Disease (ESRD) Cohort involving approximately 60 subjects undergoing maintenance hemodialysis at Severance Hospital in Seoul, Korea. The subset of participants subjected to analysis consisted of stable individuals who provided informed consent and had not undergone hospitalization for reasons related to infection or acute events within the preceding three months.

(11) There are no data on the purity of IS.

According to the reviewer's suggestion, we have included information regarding the purity (99%) of IS in the Methods section.

(12) Figure 6L: Immunoblot on b-actin were merged. This reviewer wonders how the authors analyzed these blots.

We express gratitude for the constructive criticism provided by the reviewer, and we acknowledge and comprehend the concerns raised. In response to the reviewer's comments, a reanalysis of the ALOX5 expression level in Figure 6M was conducted, employing immunoblot analysis on β-actin, as depicted in Figure 6L, with a short exposure time (Author response image 22).

**Author response image 22. sa2fig22:** ALOX5 protein exhibited an elevation in splenic myeloid cells obtained from IS-trained mice.

(13) qPCR data throughout the manuscript have control group with no error bar. The authors may not set all controls arbitrarily equal to 1 (Example Figure 1H and I). Data should be normalized in a test standard way. The average of a single datapoint may be scaled to 1, but variation must remain within the control groups.

We express gratitude to the reviewer for their valuable feedback, acknowledging a comprehensive understanding of their perspectives. Our qPCR assays predominantly investigated the impact of various treatments on the expression of specific target genes (e.g., TNF-α, IL-6, Alox5) within monocytes/macrophages obtained from the same donors.

Subsequently, normalization of gene expression levels occurred relative to ACTINB expression, followed by relative fold-increase determination using the comparative CT method (ΔΔCT).

Statistical significance was assessed through a two-tailed paired analysis in these instances. Additionally, a substantial portion of the qPCR data was validated at the protein level through ELISA and immunoblotting techniques.

Minor Comments:(1) Molecular weight markers are missing in immunoblots throughout the manuscript.

According to the reviewer's comment, molecular weight markers are added into immunoblots

(2) ESRD should be spelled out in the title.

According to the reviewer's comment, we spelled out ESRD in the title.